# Conjugated Semantic Pool Improves OOD Detection with Pre-trained Vision-Language Models

**Mengyuan Chen**
MAIS, Institute of Automation, CAS
School of Artificial Intelligence, UCAS
`chenmengyuan2021@ia.ac.cn`

**Junyu Gao**[*]
MAIS, Institute of Automation, CAS
School of Artificial Intelligence, UCAS
`junyu.gao@nlpr.ia.ac.cn`

**Changsheng Xu**[*]
MAIS, Institute of Automation, CAS
School of Artificial Intelligence, UCAS
Pengcheng Laboratory
`csxu@nlpr.ia.ac.cn`

## Abstract

A straightforward pipeline for zero-shot out-of-distribution (OOD) detection involves selecting potential OOD labels from an extensive semantic pool and then leveraging a pre-trained vision-language model to perform classification on both in-distribution (ID) and OOD labels. In this paper, we theorize that enhancing performance requires expanding the semantic pool, while increasing the expected probability of selected OOD labels being activated by OOD samples, and ensuring low mutual dependence among the activations of these OOD labels. A natural expansion manner is to adopt a larger lexicon; however, the inevitable introduction of numerous synonyms and uncommon words fails to meet the above requirements, indicating that viable expansion manners move beyond merely selecting words from a lexicon. Since OOD detection aims to correctly classify input images into ID/OOD class groups, we can "make up" OOD label candidates which are not standard class names but beneficial for the process. Observing that the original semantic pool is comprised of unmodified specific class names, we correspondingly construct a conjugated semantic pool (CSP) consisting of modified superclass names, each serving as a cluster center for samples sharing similar properties across different categories. Consistent with our established theory, expanding OOD label candidates with the CSP satisfies the requirements and outperforms existing works by 7.89% in FPR95. Codes are available in https://github.com/MengyuanChen21/NeurIPS2024-CSP.

## 1 Introduction

The efficacy of machine learning models typically diminishes on out-of-distribution (OOD) data, thereby underscoring the significance of flagging OOD samples for caution. Traditional visual OOD detection methods are typically driven by a single image modality, leaving the rich information in textual labels untapped [21, 34, 63, 56, 11]. As pre-trained vision-language models (VLMs) develop, employing textual information in visual OOD detection has become a burgeoning paradigm [15, 12, 40, 58, 64, 43, 29]. A straightforward pipeline [29] is to select potential OOD labels from a semantic pool and leverages the text-image alignment ability of a pre-trained VLM. Specifically, potential OOD labels are selected from WordNet [39] based on their similarities to the In-distribution (ID) label space, and then CLIP [47] is employed to classify input images into ID/OOD class groups.

---

[*]Corresponding authors

38th Conference on Neural Information Processing Systems (NeurIPS 2024).

In this paper, we establish a mathematic model to describe the performance of the above pipeline. Specifically, the activation status of selected OOD labels, aka whether the similarities between OOD labels and an input image exceed an implicit threshold, can be modeled as a series of independent Bernoulli random variables [29]. Depending on the class group of the input image, we refer these as ID and OOD Bernoulli variables for short. As the proportion of selected OOD labels in the semantic pool increases, our theory indicates an inverted-V performance variation trend, which aligns with the actual observation. We further derive that the peak performance is positively correlated with two factors: the size of the semantic pool and the average expectation of the OOD Bernoulli variables. Considering the mutual influence between factors, a clear strategy for enhancing performance involves concurrently enlarging these two factors while maintaining low mutual dependence among the Bernoulli variables. As a result, with an existing semantic pool, what we need to do is to *expand it with additional OOD labels which have higher and independent probabilities of being activated by OOD images*.

A straightforward manner of semantic pool expansion is to adopt larger lexicons. However, simple lexicon expansion fails to yield consistent satisfactory outcomes, and we conclude that the inefficacy is attributed to the following reasons: On the one hand, larger lexicons bring numerous uncommon words, whose expected probabilities of being activated by OOD images are minimal, thus resulting in a reduction of the average expectation of the OOD Bernoulli variables. On the other hand, larger lexicons introduce plenty of (near-)synonyms for existing OOD label candidates, leading to a high degree of functional overlap and little benefit. The corresponding Bernoulli random variables for (near-)synonyms are highly mutual dependent, which severely violates the independence assumption required by Lyapunov central limit theorem [1], thus failing to achieve the expected enhancement.

The above analysis suggests that viable strategies for semantic pool expansion require moving beyond the paradigm of simply selecting labels from larger lexicons. Since the goal of OOD detection is to correctly classify input images into ID/OOD class groups, we can freely "make up" OOD label candidates which are not standard class names but beneficial for the process. Inspired by the fact that the original semantic pool is comprised of **unmodified specific class names** (*e.g.*, *"cat"*, *"wallet"*, *"barbershop"*), each of which serves as a cluster center for samples from the same category but with varying properties, we correspondingly construct a conjugated semantic pool (CSP) consisting of **specifically modified superclass names** (*e.g.*, *"white creature"*, *"valuable item"*, *"communal place"*), each of which serves as a cluster center for samples sharing similar properties across different categories. Expanding OOD label candidates with the CSP satisfies the requirements of our theoretical scheme. Specifically, since superclasses used in constructing the CSP include broad semantic objects, the property clusters encompass samples from numerous potential OOD categories. Therefore, these cluster centers, serving as OOD labels, have much higher expected probabilities of being activated by OOD samples, thus increasing the average expectation of the OOD Bernoulli variables. Furthermore, the distribution of these property cluster centers in the feature space is distinctly different from that of the original category cluster centers, resulting in a relatively low mutual dependence between the new and original labels. Consistent with the established theory, our method outperforms the SOTA method NegLabel [29] with an improvement of 7.89% in FPR95, which underscores the efficacy of our method. Our contributions include:

- A theoretical scheme for improving OOD detection with pre-trained VLMs (Section 3.1). We derive that an unequivocal strategy for performance enhancement requires concurrently increasing the semantic pool size and the expected activation probability of OOD labels and ensuring low mutual dependence among the activations of selected OOD labels.

- An analysis of the inefficacy of simple lexicon expansion (Section 3.2). We attribute the inefficacy to the introduction of numerous uncommon words and (near-)synonyms, which respectively reduces the expected activation probabilities of OOD labels and brings severe mutual dependence, thereby failing to achieve theoretical enhancement.

- An expansion manner beyond selecting labels from existing lexicons (Section 3.3). We construct an additional conjugated semantic pool (CSP), consisting of modified superclass names, each serving as a cluster center for samples with similar properties across different categories. Consistent with our established theory, expanding OOD label candidates with the CSP satisfies the requirements and achieves satisfactory performance improvements.

- Extensive experiments and related analysis on multiple OOD detection benchmarks with state-of-the-art performances (Section 5), which demonstrate the effectiveness of our method.

Proof and derivations, visualizations, additional experiment results and details are given in Appendix.

## 2 Preliminaries

**Task setup.** OOD detection leveraging pre-trained vision-language models (VLMs), also termed as zero-shot OOD detection [15, 12, 40, 58, 64, 43, 29], aims to identify OOD images from ID ones with only natural-language labels of ID classes available. Formally, given the testing image set $\mathcal{X} = \mathcal{X}^{\text{in}} \cup \mathcal{X}^{\text{out}}$, where $\mathcal{X}^{\text{in}} \cap \mathcal{X}^{\text{out}} = \emptyset$, and ID label (class name) set $\mathcal{Y}^{\text{in}} = \{y_1, \ldots, y_K\}$, where $K$ is the number of ID classes, our target is to obtain an OOD detector $G(x; \mathcal{Y}^{\text{in}}) : \mathcal{X} \to \{\text{in}, \text{out}\}$, where $x \in \mathcal{X}$ denotes a test image. It is noteworthy that the zero-shot setting does not require that there be no overlap between the pre-training data of VLMs and the testing data $\mathcal{X}$, but only stipulates that no ID images are available for model fine-tuning. In other words, the split of ID and OOD data completely depends on how users manually preset the ID label set $\mathcal{Y}_{\text{in}}$.

**OOD detection with a pre-trained VLM and a semantic pool.** A straightforward pipeline of this task is to select potential OOD labels from a semantic pool and leverages the text-image alignment ability of a pre-trained VLM to perform zero-shot OOD detection [29]. Specifically, there are three steps: (1) Fetching numerous words from a semantic pool like WordNet [39] as OOD label candidates; (2) Selecting a portion of OOD label candidates most dissimilar to the entire ID label space; (3) Employing a pre-trained VLM like CLIP [47] to obtain similarities between testing images and ID/OOD labels and then performing OOD detection with a designed OOD score.

The OOD detection performance of this pipeline can be modeled as follows [29]. Given the selected OOD label set $\mathcal{Y}^{\text{out}} = \{z_1, \ldots, z_m\}$, $0 < m \leq M$, where $m$ is the number of selected OOD labels, and $M$ is the size of the semantic pool. By applying a threshold $\psi$, we can naturally define $p_i^{\text{in}} = P(s_i \geq \psi | f, z_i, \mathcal{X}^{\text{in}})$ as the probability of classifying ID input images $x \in \mathcal{X}^{\text{in}}$ as positive for the given label $z_i$, where $s_i = \text{sim}(f(x), f(z_i))$ is the similarity score given by the pre-trained model $f$. To derive an analytic form for the model's OOD detection performance, we employ a straightforward OOD score function $S(x)$, aka the total positive count across categories for a sample $x$. Specifically, $S(x^{\text{in}}) = \sum_i s_i^{\text{in}}$, where $s_i^{\text{in}}$ is a Bernoulli random variable with parameter $p_i^{\text{in}}$, *i.e.*, the probability of $s_i^{\text{in}} = 1$ is $p_i^{\text{in}}$ and the probability of $s_i^{\text{in}} = 0$ is $1 - p_i^{\text{in}}$. Consequently, $S(x^{\text{in}})$ follows a Poisson binomial distribution with parameters $\{p_1^{\text{in}}, ..., p_m^{\text{in}}\}$. $p_i^{\text{out}}$ and $S(x^{\text{out}})$ are defined similarly. Based on the Lyapunov central limit theorem (CLT) [1], we can obtain the following lemma:

**Lemma 1.** *Given independent Bernoulli random variables $\{s_1, ..., s_m\}$ with parameters $\{p_1, ..., p_m\}$, where $0 < p_i < 1$, as $m$ goes to infinity, the Poisson binomial random variable $C = \sum_{i=1}^{m} s_i$ converges in distribution to a normal random variable with distribution $\mathcal{N}\left(\sum_{i=1}^{m} p_i, \sum_{i=1}^{m} p_i(1 - p_i)\right)$.*

According to Lemma 1, proved in Appendix A.1, the distribution of $C^{\text{in}}$ can be approximated as $C^{\text{in}} \sim \mathcal{N}\left(\sum_{i=1}^{m} p_i^{\text{in}}, \sum_{i=1}^{m} p_i^{\text{in}}(1 - p_i^{\text{in}})\right)$, and the distribution of $C^{\text{out}}$ can be approximated similarly. By denoting $q_1 = \mathbb{E}_i[p_i^{\text{in}}]$, $v_1 = \text{Var}_i[p_i^{\text{in}}]$, $q_2 = \mathbb{E}_i[p_i^{\text{out}}]$, $v_2 = \text{Var}_i[p_i^{\text{out}}]$, we have

$$C^{\text{in}} \sim \mathcal{N}\left(mq_1, mq_1(1 - q_1) - mv_1\right), C^{\text{out}} \sim \mathcal{N}\left(mq_2, mq_2(1 - q_2) - mv_2\right). \quad (1)$$

Thereafter, with the derivation provided in Appendix A.2, we can obtain the closed-form expression of one of the most commonly adopted OOD performance metric, aka the false positive rate (FPR) when the true positive rate (TPR) is $\lambda \in [0, 1]$, denoted by $\text{FPR}_\lambda$, as

$$\text{FPR}_\lambda = \frac{1}{2} + \frac{1}{2} \cdot \text{erf}\left(\sqrt{\frac{q_1(1 - q_1) - v_1}{q_2(1 - q_2) - v_2}} \text{erf}^{-1}(2\lambda - 1) + \frac{\sqrt{m}(q_1 - q_2)}{\sqrt{2q_2(1 - q_2) - 2v_2}}\right), \quad (2)$$

where $\text{erf}(x) = \frac{2}{\sqrt{\pi}} \int_0^x e^{-t^2} dt$. However, contrary to the monotonic trend suggested by Eqn. 2, the actual performances in experiments exhibit an inverted-V trend as the ratio of selected OOD labels in the semantic pool increases. Therefore, we further optimize the mathematic model by incorporating finer-grained variable relationships, seeking theoretical guidance for performance enhancement.

## 3 Methodology

### 3.1 A Theoretical Scheme for Performance Enhancement

Since selecting OOD labels is typically based on the reverse-order of similarities to the ID label space to minimize semantic overlap, *i.e.*, the most dissimilar OOD label candidates are most likely

to be selected, the expected probability, $q_1$, of OOD labels being activated by ID images is not static. Specifically, as the ratio of selected OOD labels $r = m/M$ increases, the expected activation probability $q_1 = \mathbb{E}_i[p_i^{\text{in}}]$ of existing OOD labels for ID images will monotonically increase, since more OOD labels with higher affinities to ID labels are selected. When all labels in the semantic pool are finally selected, $q_1$ will achieve $q_2$, which means the expected probabilities of OOD labels being activated by ID and OOD images are close. Meanwhile, $q_2$ is considered as a constant when the semantic pool is fixed and the ratio $r$ varies, since whether an element in the pool (excluding ID labels) corresponds to a potential OOD sample is independent of its similarity to the ID label space. Formally, defining $q_0$ as the lower bound of $q_1$, we model the accumulated increase in $q_1$ as the ratio $r$ increases from zero with the function $u(r)$, which exhibits following properties:

$$u(r) = q_1(r) - q_0, \ u(r = 0|q_0, q_2) = 0, \ u(r = 1|q_0, q_2) = q_2 - q_0 > 0, \ u'(r) \geq 0. \quad (3)$$

Besides, we assume that the absolute value of the curvature of $u$ is constrained within a specific range, thereby preventing abrupt changes in the trend of $u$, which facilitates subsequent analysis. With $u(r)$, we set $\lambda = 0.5$ in Eqn. 2 for convenience and then explore the properties of

$$\text{FPR}_{0.5} = \frac{1}{2} + \frac{1}{2} \cdot \text{erf}\left(\sqrt{\frac{m}{2}} \cdot \frac{q_0 - q_2 + u(r|q_0, q_2)}{\sqrt{q_2(1 - q_2) - v_2}}\right). \quad (4)$$

Denote $z = \sqrt{\frac{m}{2}} \cdot \frac{q_0 - q_2 + u}{\sqrt{q_2(1-q_2) - v_2}}$, from Eqn. 4, we can derive that the first-order derivative of $\text{FPR}_{0.5}$, denoted as $G(r)$, can be expressed as

$$G(r|q_0, q_2, u, M) = \frac{\partial \text{FPR}_{0.5}}{\partial r} = \frac{M e^{-z^2}}{2\sqrt{2\pi}} \cdot \frac{q_0 - q_2 + u + 2ru'}{\sqrt{m}\sqrt{q_2(1 - q_2) - v_2}}, \quad (5)$$

which can be further proved to monotonically increase over the interval $(0, 1]$ with respect to $r$ with the above assumptions. Besides, according to Eqn. 5, we can obtain that

$$\lim_{r \to 0^+} G(r) = \lim_{r \to 0^+} \frac{\kappa(q_0 - q_2)}{2\sqrt{r}} = -\infty, \quad \lim_{r \to 1} G(r) = \kappa u'(r = 1) \geq 0, \quad (6)$$

where $\kappa = (M/2\pi)^{\frac{1}{2}}(q_2(1 - q_2) - v_2)^{-\frac{1}{2}} e^{-z^2} > 0$.

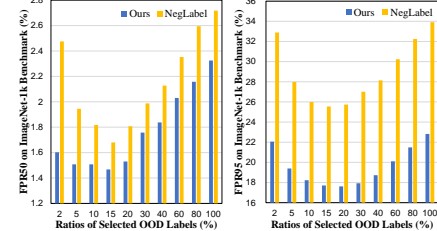

Since $\text{FPR}_{0.5}(r)$ is a continuous function within our framework and satisfies Eqn. 6, it can be deduced that there exists a value $r_0 \in (0, 1]$ where $\text{FPR}_{0.5}$ reaches its minimum. Furthermore, $\text{FPR}_{0.5}$ monotonically decreases over the interval $(0, r_0]$ and increases over the interval $[r_0, 1]$. Since $\text{FPR}_\lambda$ is a smooth continuous function with respect to $\lambda$, we deduce that $\text{FPR}_\lambda$ and $\text{FPR}_{0.5}$ share similar trends as the parameter $r$ varies, which aligns with the actual results presented in Fig. 1. A more detailed calculation process from Eqn. 4 to Eqn. 6 is provided in Appendix A.3.

Figure 1: Model performances evaluated by FPR50 and FPR95 (lower is better) of our method and NegLabel against the ratio $r$, which exhibit a trend of initial decline followed by an increase. Detailed results can be found in Table 8.

Subsequently, we delve deeper into the factors influencing the optimal value of the OOD detection performance evaluated by $\text{FPR}_{0.5}(r)$. When $r$ reaches the critical point $r_0$, the expected performance improvement resulting from "OOD samples being correctly identified due to the addition of new OOD labels" will be equal to the performance degradation caused by "ID samples being misclassified due to the addition of new OOD labels". As a result, the model performance achieves its peak. Specifically, from Eqn. 5, it can be inferred that $r_0$ satisfies

$$q_0 - q_2 + u(r_0|q_0, q_2) + 2r_0 u'(r_0|q_0, q_2) = 0. \quad (7)$$

Given the complex interdependencies among the variables in the above equation, it is challenging to derive any definitive conclusions with the undefined form of the function $u$. Consequently, we simplify by assuming that the function $u(r)$ is linear. Under this assumption, by substituting Eqn. 7 into Eqn. 4, we obtain that the optimal value of $\text{FPR}_{0.5}$ can be expressed as

$$\text{FPR}_{0.5}(r_0) = \frac{1}{2} + \frac{1}{2} \cdot \text{erf}\left(-\frac{(2M)^{\frac{1}{2}} r_0^{\frac{3}{2}}(q_2 - q_0)}{\sqrt{q_2(1 - q_2) - v_2}}\right), \quad (8)$$

where the variables $M$ and $q_2$, aka the semantic pool size and the expected probability of OOD labels being activated by OOD samples, are the predominant factors influencing the right side of the equation. The other variables $q_0$, $r_0$, and $v_2$ remain nearly constant with a sufficiently large semantic pool (refer to Appendix A.4 for analysis), thus exerting marginal impact.

If we disregard the interdependencies among the variables, the impact of $M$ and $q_2$ on the optimal value of $\text{FPR}_{0.5}$ is straightforward: (1) With $q_2$ fixed, it can be easily observed that $\text{FPR}_{0.5}(r_0)$ monotonically decreases with $M$. (2) With $M$ fixed, and denoting the input to the $\text{erf}(\cdot)$ function as $\zeta$, it can be derived from Eqn. 8 that,

$$\frac{\partial \text{FPR}_{0.5}(r_0)}{\partial q_2} = \frac{1}{2} \cdot \frac{\partial \text{erf}(\zeta)}{\partial \zeta} \cdot \frac{\partial \zeta}{\partial q_2} = -\sqrt{\frac{Mr_0^3}{2\pi}} \cdot \frac{e^{-\zeta^2}(q_2 + q_0 - 2q_0q_2 - 2v_2)}{(q_2(1-q_2) - v_2)^{\frac{3}{2}}} \leq 0 \quad (9)$$

holds in almost all practical cases (see Appendix A.5 for analysis), thus $\text{FPR}_{0.5}(r_0)$ also monotonically decreases with respect to $q_2$. However, in real-world scenarios, the complex interactions between $M$ and $q_2$ prevent either variable from being adjusted in isolation. For instance, utilizing a larger lexicon to expand the size $M$ of the semantic pool may cause a decline in $q_2$ (see Section 3.2). Conversely, discarding candidates with lower activation probabilities to elevate $q_2$ leads to a reduction of $M$. The variable changes in both strategies exert opposing effects, ultimately leading to minimal improvements or even degradation in model performance. Besides, the Lyapunov central limit theorem (CLT) used in proof of Lemma 1 requires that the activations of selected OOD labels are independent. Although complete independence is impossible to achieve in real-world scenarios, it is essential to maintain a relatively low level of mutual dependence to reduce the errors in theoretical derivations. Therefore, an unequivocal strategy for performance enhancement is **concurrently increasing the variables $M$ and $q_2$ and ensuring that there is no strong dependence among the activations of selected OOD labels**. With an existing semantic pool, what we need to do is to expand it with additional OOD labels which have higher and independent probabilities of being activated by OOD images.

## 3.2 A Closer Look at the Inefficacy of Simple Lexicon Expansion

Therefore, it is time to consider how to expand the original semantic pool, which already includes most common words, while ensuring the increase of $q_2$ and low mutual dependence. The most straightforward strategy for expansion, adopting larger existing lexicons, fails to consistently yield satisfactory outcomes, as shown in Fig. 2. Subsequently, we analyze that the inefficacy of simple lexicon expansion is attributed to the following reasons.

On the one hand, larger lexicons bring numerous **uncommon words**, whose expected probability of being activated by OOD images are minimal, thus **resulting in a reduction of $q_2$**. As derived in Section 3.1, the decrease of $q_2$ attenuates the performance improvements yielded by enlarging $M$. There are two potential reasons for the activation probability $p_i^{\text{out}}$ of an uncommon OOD label $z_i$ being minimal: (1) Pre-trained VLMs lack semantic matching capability for label $z_i$. This issue is particularly pronounced when $z_i$ pertains to concepts such as highly abstract notions (e.g., *"idealism"*, *"metaphysics"*),

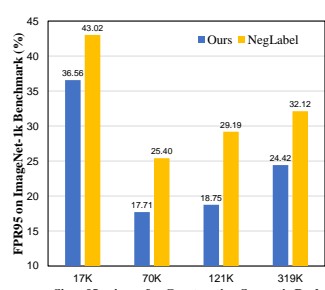

Figure 2: Model performances evaluated by FPR95 (lower is better) with lexicons of different sizes. Detailed results can be found in Table 9.

complex mathematical concepts (e.g., *"Lyapunov condition"*, *"central limit theorem"*), or specific knowledge of individuals (e.g., personal names excluding celebrities). Pre-trained VLMs are unable to recognize the corresponding content of these text inputs, resulting in $p_i^{\text{out}}$ remaining close to zero. (2) The set $\mathcal{X}^{\text{out}}$ lacks testing samples similar to label $z_i$. For instance, when the test dataset primarily consists of images of everyday items, new labels constructed from astronomical terms are likely to maintain $p_i^{\text{out}}$ close to zero. [2] The above scenarios are much more prevalent in lexicons of uncommon terms than those of common words. Thereafter, a larger proportion of labels with minimal activation probability $p_i^{\text{out}}$ will diminish $q_2 = \mathbb{E}_i[p_i^{\text{out}}]$, thus attenuating the performance improvement.

On the other hand, larger lexicons introduce plenty of **synonyms and near-synonyms** for existing OOD label candidates, leading to a high degree of functional overlap with little additional benefit.

---

[2]In practical applications, input images typically encompass limited categories. Therefore, it is reasonable that an infinite expansion of vocabulary richness does not necessarily yield sustained performance improvement.

For example, the common word *"smartphone"* can be expanded by adding synonyms such as *"mobile phone"* and *"cellphone"*. However, the selection results for these words are consistent due to their similar meanings, and if they are selected, the activation of these labels still depends solely on the presence of a smartphone in the input image. This demonstrates a high level of mutual dependency and provides little additional benefit compared to only including *"smartphone"* in the semantic pool. Despite no reduction in $q_2$, the corresponding Bernoulli random variables for synonyms, representing whether they are activated by an input OOD image, severely **violate the independent assumption** required by Lemma 1. Although the Lyapunov CLT used in proof of Lemma 1 relaxes the requirement for random variables to have strictly identical distributions as mandated by the traditional CLT, it still requires that the variables maintain mutual independence. Despite the random variables corresponding to semantically dissimilar labels are not strictly independent, the intensity of their mutual dependency is generally much lower than that observed in (near-)synonyms. Contrarily, synonyms and near-synonyms lead to significant bias in the approximation of Eqn. 1 and the conclusions derived, thereby failing to achieve the theoretical enhancement.

### 3.3 Expanding Label Candidates with Conjugated Semantic Pool

The above analysis suggests that viable strategies for semantic pool expansion require moving beyond the paradigm of simply selecting labels from a lexicon. Since the goal of OOD detection is to correctly classify input images into ID/OOD class groups, we can freely "make up" OOD label candidates which are not standard class names but beneficial for the process. Inspired by the fact that the original semantic pool is comprised of **unmodified specific class names**, each of which serves as a cluster center for samples from the same category but with varying properties, we correspondingly construct a conjugated semantic pool (CSP) consisting of **specifically modified superclass names**, each of which serves as a cluster center for samples sharing similar properties across different categories.

We notice that a class name inherently encompasses a broad semantic range. As shown in the bottom right of Fig. 3, when an image is attached with the class label *"cat"*, it actually depicts one of various more specific situations, such as a *"white cat"*, *"tabby kitten"*, *"gray cat"*, *"yawning cat"*, or *"cat on a mat"*, etc. Considering all feature points that correspond to more specific descriptions of cats as a cluster within the feature space, the feature point of *"cat"* can be regarded as its cluster center.

In an ideal scenario, each input image in the feature space would be closest (most similar) to the cluster center that corresponds to its category, thereby achieving perfect OOD identification. However, the following issues impair the ideal case: (1) Due to the limited capabilities of pre-trained VLMs, some OOD images, such as *"white polar bear"* in Fig. 3, are closer to incorrect cluster centers than to the correct ones. (2) Owing to the limited scope of lexicons and the inaccuracy of label selection, the category name corresponding to an OOD image, such as *"white peacock"* in Fig. 3, may not exist in selected labels, resulting in the absence of an appropriate cluster center. To summarize briefly: not every input OOD image locates close to a correct OOD cluster center.

Thereafter, it naturally occurs to us that we should construct more suitable cluster centers to attract such

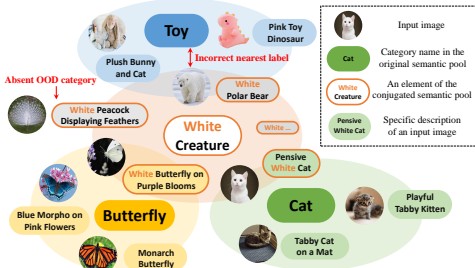

Figure 3: An illustrative diagram of an element in the conjugated semantic pool (CSP). Category names can be regarded as the centers of category clusters. Similarly, elements in CSP can be considered as cluster centers of superclass objects with similar properties.

"homeless" OOD images. This is why we expand the original semantic pool by constructing the CSP as follows: Instead of specifying concrete category names (*e.g.*, *"cat"*, *"wallet"*, *"barbershop"*), we utilize superclass names to encompass a wider range of categories (*e.g.*, *"creature"*, *"item"*, *"place"*); Instead of leaving category names undecorated, we using adjectives from a lexicon as modifiers to attract objects sharing similar properties. As a result, we obtain numerous label candidates of random combinations of adjectives and superclasses (*e.g.*, *"white creature"*, *"valuable item"*, *"communal place"*). As Fig. 3 shows, in the feature space, *"white creature"* can be considered as the cluster center of all feature points corresponding to creatures modified by *"white"*, such as *"white cat"*, *"white butterfly"*, *"white polar bear"*, etc. Note that the semantic scopes of label candidates in the CSP may

overlap with ID categories. For example, when *"Cat"* or *"Butterfly"* in Fig. 3 are included in ID classes, the label candidate *"White Creature"* in the CSP may not be selected as an OOD label.

Consistent with our established theory, our proposed method achieves satisfactory performance improvements by concurrently enlarging the semantic pool size $M$ and the expected activation probability $q_2$ of OOD labels and ensuring that there is no severe mutual dependence among the activations of selected OOD labels. Firstly, when we expand the original semantic pool with the CSP, the enlargement of $M$ is obvious. Then, since the superclasses used in constructing the CSP typically include broad semantic objects, the property clusters encompass samples from numerous potential OOD categories. Therefore, their centers have much higher expected probabilities of being activated by OOD samples, which brings an increase in $q_2$. Furthermore, the distribution of these property cluster centers in the feature space is distinctly different from that of the original category cluster centers. As a result, the mutual dependence between the new and original labels is relatively low, and the functions of labels from the CSP will not be overshadowed, enhancing the likelihood that an OOD image locates close to a correct OOD cluster center. Experiment results and analysis which support the above claims are provided in Appendix C.1.

## 4    Related works

**Traditional visual OOD detection.** Traditional visual OOD detection methods, driven by the single image modality, can be broadly categorized into four distinct types: (1) Output-based methods, which aims to obtain improved OOD scores from network output, can be further classified into post-hoc methods [21, 34, 25, 54, 55, 44, 38] and training-based ones [10, 23, 57, 26, 71, 65, 30]. (2) Density-based methods [37, 49, 53, 67, 11] explicitly model the ID data with probabilistic models and identify test data located in regions of low density as OOD. (3) Distance-based methods [32, 51, 63, 56, 41, 7, 70, 59, 24, 18] originate from the core idea that OOD samples should be relatively far away from ID prototypes or centroids. (4) Reconstruction-based methods [76, 69, 28, 33], which employ an encoder-decoder framework trained on ID data, leverage the performance discrepancies between ID and OOD samples as indicators for anomaly detection. Furthermore, numerous studies [52, 72, 14, 42, 6] offer theoretical contributions.

**OOD detection leveraging pre-trained VLMs.** By adopting pre-trained VLMs, employing textual information in visual OOD detection has become a burgeoning paradigm with remarkable performance [15, 12, 40, 58, 64, 43, 29]. Fort *et al.*[15] propose to feed the names of potential outlier classes to image-text pre-trained transformers like CLIP [47] for OOD detection. ZOC [12] extends CLIP with a text-based image description generator to output OOD label candidates for testing. MCM [40] simply adopts maximum predicted softmax value as the OOD score, which is an effective and representative post-hoc OOD detection method based on vision-language pre-training. Based on MCM, NPOS [58] generates artificial OOD training data and facilitates learning a reliable decision boundary between ID and OOD data. CLIPN [64] trains a text encoder to teach CLIP to comprehend negative prompts, effectively discriminating OOD samples through the similarity discrepancies between two text encoders and the frozen image encoder. Also based on CLIP, LSN [43] constructs negative classifiers by learning negative prompts to identify images not belonging to a given category. NegLabel [29] proposes a straightforward pipeline, that is, selecting potential OOD labels from an extensive semantic pool like WordNet [39], and then leveraging a pre-trained VLM like CLIP to classify input images into ID/OOD class groups. In this study, we explore the theoretical requirements for performance enhancement in this pipeline, and thus construct a conjugated semantic pool to expand OOD label candidates, which achieves performances improvements as expected.

**Further discussion about NegLabel.** NegLabel [29] undertakes a rudimentary theoretical analysis of the correlation between OOD detection performance and the quantity of adopted potential labels, concluding that an increase in selected labels correlates with enhanced performance. However, this conclusion contradicts the observed actual trend. The contradiction arises from that [29] simply assume a constant higher similarity between OOD labels and OOD images compared to ID images, neglecting that this similarity discrepancy originates from the strategy of reverse-order selection of OOD labels based on their similarity to the ID label space. As the set of selected OOD labels transitions from "*a small subset of labels with the lowest similarity to the entire ID label space*" to "*the whole semantic pool, which is unrelated to the setting of ID and OOD labels*", the discrepancy in similarity of ID images to OOD labels versus OOD images to OOD labels will progressively diminish until it disappears. Incorporating the above dynamic to optimize the mathematic model, we focus

on the correlation between OOD detection performance and the ratio of selected OOD labels in the semantic pool, seeking theoretical guidance for performance enhancement.

## 5 Experiments

### 5.1 Experiment Setup

**Benchmarks.** We mainly evaluate our method on the widely-used ImageNet-1k OOD detection benchmark [26]. This benchmark utilizes the large-scale ImageNet-1k dataset as the ID data, and select samples from iNaturalist [60], SUN [66], Places [73], and Textures [8] as the OOD data. The categories of the OOD data have been manually selected to prevent overlap with ImageNet-1k. Furthermore, we conduct experiments on hard OOD detection tasks, or with various ID datasets. Besides, we access whether our method generalizes well to different VLM architectures, including ALIGN [27], GroupViT [68], EVA [13], etc. More details of datasets can be found in Appendix B.

**Implementation details.** Unless otherwise specified, we employ the CLIP ViT-B/16 model as the pre-trained VLM and WordNet as the lexicon. The superclass set for constructing the conjugated semantic pool is {*area, creature, environment, item, landscape, object, pattern, place, scene, space, structure, thing, view, vista*}, which nearly encompasses all real-world objects. The ablation in Appendix C.5 shows that numerous alternative selections can also yield significant performance improvements. All hyper-parameters are directly inherited from [29] without any modification, including the ratio $r$ which is set to $15\%$. Additionally, we adopt the same NegMining algorithm, OOD score calculation method, and grouping strategy as described in [29]. All experiments are conducted using GeForce RTX 3090 GPUs.

**Prompt ensemble.** We use the following prefixes to construct prompts for labels in the original semantic pool: *the, the good (nice), a photo of (with) the nice, a good (close-up) photo of the nice*. For labels in the conjugated semantic pool, we apply the prefixes: *a nice (good, close-up) photo of*. The baseline results reported in the ablation study (see Table 3) also utilize this technique.

**Computational cost.** The prompt ensemble is constructed over the embedding space to avoid any additional inference cost. Similar to NegLabel, our method is a post hoc OOD detector with negligible extra computational burden, which introduces $< 1\%$ network forward latency.

**Evaluation metrics.** Following previous works [40, 29, 64], we adopt the following metrics: the area under the receiver operating characteristic curve (AUROC), and the false positive rate of OOD data when the true positive rate of ID data is 95% (FPR95) [46].

### 5.2 Evaluation on OOD detection benchmarks

**Evaluation on ImageNet-1k OOD detection benchmark.** We compare our method with existing OOD detection methods on the ImageNet-1k benchmark organized by [26] in Table 1. The methods

Table 1: Comparative performance of OOD detection across baseline methods utilizing CLIP ViT-B/16 architecture with ImageNet-1k as ID data. Performance metrics are presented as percentages.

| Methods | OOD Datasets | | | | | | | | Average | |
| | iNaturalist | | SUN | | Places | | Textures | | | |
| | AUROC↑ | FPR95↓ | AUROC↑ | FPR95↓ | AUROC↑ | FPR95↓ | AUROC↑ | FPR95↓ | AUROC↑ | FPR95↓ |
|---|---|---|---|---|---|---|---|---|---|---|
| MSP [21] | 87.44 | 58.36 | 79.73 | 73.72 | 79.67 | 74.41 | 79.69 | 71.93 | 81.63 | 69.61 |
| ODIN [34] | 94.65 | 30.22 | 87.17 | 54.04 | 85.54 | 55.06 | 87.85 | 51.67 | 88.80 | 47.75 |
| Energy [37] | 95.33 | 26.12 | 92.66 | 35.97 | 91.41 | 39.87 | 86.76 | 57.61 | 91.54 | 39.89 |
| GradNorm [25] | 72.56 | 81.50 | 72.86 | 82.00 | 73.70 | 80.41 | 70.26 | 79.36 | 72.35 | 80.82 |
| ViM [63] | 93.16 | 32.19 | 87.19 | 54.01 | 83.75 | 60.67 | 87.18 | 53.94 | 87.82 | 50.20 |
| KNN [56] | 94.52 | 29.17 | 92.67 | 35.62 | 91.02 | 39.61 | 85.67 | 64.35 | 90.97 | 42.19 |
| VOS [11] | 94.62 | 28.99 | 92.57 | 36.88 | 91.23 | 38.39 | 86.33 | 61.02 | 91.19 | 41.32 |
| ZOC [12] | 86.09 | 87.30 | 81.20 | 81.51 | 83.39 | 73.06 | 76.46 | 98.90 | 81.79 | 85.19 |
| MCM [40] | 94.59 | 32.20 | 92.25 | 38.80 | 90.31 | 46.20 | 86.12 | 58.50 | 90.82 | 43.93 |
| NPOS [58] | 96.19 | 16.58 | 90.44 | 43.77 | 89.44 | 45.27 | 88.80 | 46.12 | 91.22 | 37.93 |
| CoOp [75] | 94.89 | 29.47 | 93.36 | 31.34 | 90.07 | 40.28 | 87.58 | 54.25 | 91.47 | 38.83 |
| CoCoOp [74] | 94.73 | 30.74 | 93.15 | 31.18 | 90.63 | 38.75 | 87.92 | 53.84 | 91.61 | 38.63 |
| CLIPN [64] | 95.27 | 23.94 | 93.93 | 26.17 | 92.28 | 33.45 | 90.93 | 40.83 | 93.10 | 31.10 |
| LSN [43] | 95.83 | 21.56 | 94.35 | 26.32 | 91.25 | 34.48 | 90.42 | 38.54 | 92.96 | 30.22 |
| NegLabel [29] | 99.49 | 1.91 | 95.49 | 20.53 | 91.64 | 35.59 | 90.22 | 43.56 | 94.21 | 25.40 |
| Ours | **99.60** | **1.54** | **96.66** | **13.66** | **92.90** | **29.32** | **93.86** | **25.52** | **95.76** | **17.51** |

Table 2: OOD detection performance comparison on hard OOD detection tasks.

| ID datasets
OOD datasets | ImageNet-10
ImageNet-20 | | ImageNet-20
ImageNet-10 | | ImageNet-10
ImageNet-100 | | ImageNet-100
ImageNet-10 | | ImageNet-1k
ImageNet-O | | WaterBirds
Placesbg | |
|---|---|---|---|---|---|---|---|---|---|---|---|---|
| Methods | AUROC | FPR95 | AUROC | FPR95 | AUROC | FPR95 | AUROC | FPR95 | AUROC | FPR95 | AUROC | FPR95 |
| MCM | 98.60 | 6.00 | 98.09 | 13.04 | 99.39 | 2.50 | 87.20 | 60.00 | 78.59 | 64.27 | 87.45 | 33.62 |
| NegLabel | 98.80 | 5.00 | 98.04 | 11.60 | 99.37 | 2.50 | 87.93 | 49.40 | 85.78 | 56.65 | 87.99 | 29.16 |
| Ours | **99.02** | **3.30** | **98.79** | **3.40** | **99.33** | **2.22** | **89.59** | **42.40** | **88.08** | **51.50** | **92.88** | **12.07** |

Table 3: Ablation study with CLIP (ViT-B/16) as the backbone on ImageNet-1k as ID.

| Components of the Whole Semantic Pool | | | OOD Datasets | | | | | | | | Average | |
|---|---|---|---|---|---|---|---|---|---|---|---|---|
| | | | iNaturalist | | SUN | | Places | | Textures | | | |
| Original
Semantic Pool | Simple
Adj Labels | Conjugated
Semantic Pool | AUROC | FPR95 | AUROC | FPR95 | AUROC | FPR95 | AUROC | FPR95 | AUROC | FPR95 |
| ✓ | | | **99.63** | **1.40** | 96.25 | 16.63 | 92.02 | 33.90 | 86.91 | 57.62 | 93.70 | 27.39 |
| | ✓ | | 97.06 | 14.00 | 93.52 | 31.82 | 90.80 | 40.23 | 90.93 | 40.89 | 93.08 | 31.74 |
| | | ✓ | 97.05 | 13.94 | 95.96 | 17.58 | 91.66 | 32.43 | **95.74** | **18.74** | 95.10 | 20.67 |
| ✓ | ✓ | | 99.60 | 1.51 | 96.15 | 16.52 | 92.38 | 32.00 | 89.26 | 47.39 | 94.35 | 24.36 |
| ✓ | | ✓ | 99.61 | 1.54 | **96.69** | **13.82** | **92.85** | **29.69** | 93.78 | 25.78 | **95.73** | **17.71** |

listed in the upper section of Table 1, ranging from MSP [21] to VOS [11], represent traditional visual OOD detection methods. Conversely, the methods in the lower section, extending from ZOC [12] to NegLabel [29], employ pre-trained VLMs like CLIP. It is evident that the integration of textual information through VLMs has increasingly become the predominant paradigm. Our method outperforms the baseline method NegLabel with a considerable improvement of 1.55% in AUROC and 7.89% in FPR95, which underscores the efficacy of our method. The reported results are averaged from runs of 10 different random seeds, whose results are provided in Appendix C.2.

**Evaluation on hard OOD detection tasks.** Following [40], we also evaluate our method on the hard OOD detection tasks in Table 2. The results of NegLabel [29] are reproduced with its released setting. Our method shows consistently high performances on various hard ID-OOD dataset pairs.

**Evaluation with various ID datasets.** We also experiment on various ID datasets, including Stanford-Cars [31], CUB-200 [61], Oxford-Pet [45], Food-101 [2], ImageNet-Sketch [62], ImageNet-A [22], ImageNet-R [20], ImageNetV2 [48], etc. Refer to Appendix C.3 for details.

### 5.3 Empirical evidence supporting our assertions

**Performance trends related to the ratio $r$.** In Fig. 1 and Table 8 (Appendix C.4), we present the FPR performances of our method and NegLabel against a progressively increasing ratio $r$, which represents the proportion of selected OOD labels in the whole semantic pool. The color gradations displayed in the table clearly illustrate an initial improvement in model performance followed by a subsequent decline as the ratio $r$ increases. This trend aligns with our derivation in Section 3.1.

**Inefficacy of simple lexicon expansion.** In Fig. 2 and Table 9 (Appendix C.4), we assess whether adopting larger lexicons enhances performances. Our findings indicate that it does not always hold. When the semantic pool covers the vast majority of common words, further expansion will introduce an excessive number of uncommon words and (near-)synonyms, thus failing to meet the derived requirements for theoretical performance enhancement. The inefficacy of simple lexicon expansion indicates that viable expansion manners move beyond merely selecting words from existing lexicons.

**Validation of consistency between methodology and theory.** Refer to Appendix C.1.

### 5.4 Ablation study

**Ablation of different semantic pool components.** As shown in Table 3, we explore the effect of three semantic pool components: (1) the original semantic pool, which consists exclusively of specific class names; (2) simple adjective labels employed by NegLabel; and (3) our conjugated semantic pool (CSP). The results demonstrate that the CSP consistently outperforms the simple adjective labels. Furthermore, the highest average performance across the four OOD datasets is achieved when the CSP is employed to expand the original semantic pool.

Table 4: Performances of OOD detection with different CLIP architectures on ImageNet-1k as ID.

| Backbones | Methods | OOD Datasets | | | | | | | | Average | |
|---|---|---|---|---|---|---|---|---|---|---|---|
| | | iNaturalist | | SUN | | Places | | Textures | | | |
| | | AUROC↑ | FPR95↓ | AUROC↑ | FPR95↓ | AUROC↑ | FPR95↓ | AUROC↑ | FPR95↓ | AUROC↑ | FPR95↓ |
| ResNet50 | NegLabel | 99.24 | 2.88 | 94.54 | 26.51 | 89.72 | 42.60 | 88.40 | 50.80 | 92.97 | 30.70 |
| | Ours | **99.46** | **1.95** | **95.73** | **19.05** | **90.39** | **38.58** | **92.41** | **32.66** | **94.50** | **23.06** |
| ResNet101 | NegLabel | 99.27 | 3.11 | 94.96 | 24.55 | 89.42 | 44.82 | 87.22 | 52.78 | 92.72 | 31.32 |
| | Ours | **99.47** | **2.04** | **95.71** | **19.50** | **90.27** | **39.57** | **90.59** | **38.67** | **94.01** | **24.95** |
| ResNet50x4 | NegLabel | 99.45 | 2.27 | 95.53 | 21.95 | 91.62 | 35.29 | 89.48 | 47.77 | 94.02 | 26.82 |
| | Ours | **99.65** | **1.48** | **96.26** | **17.02** | **92.01** | **33.42** | **92.97** | **29.40** | **95.22** | **20.33** |
| ResNet50x16 | NegLabel | 99.48 | 2.00 | 94.18 | 29.11 | 88.85 | 48.14 | 91.23 | 38.74 | 93.43 | 29.50 |
| | Ours | **99.68** | **1.25** | **95.89** | **17.89** | **91.52** | **35.77** | **93.80** | **26.61** | **95.22** | **20.38** |
| ResNet50x64 | NegLabel | 99.63 | 1.46 | 94.29 | 29.34 | 91.23 | 39.18 | 88.27 | 49.43 | 93.36 | 29.85 |
| | Ours | **99.69** | **1.19** | **96.21** | **18.49** | **92.81** | **30.52** | **92.57** | **31.12** | **95.32** | **20.33** |
| ViT-B/32 | NegLabel | 99.11 | 3.73 | 95.27 | 22.48 | 91.72 | 34.94 | 88.57 | 50.51 | 93.67 | 27.92 |
| | Ours | **99.46** | **2.37** | **96.49** | **15.01** | **92.42** | **31.47** | **93.64** | **25.09** | **95.50** | **18.49** |
| ViT-B/16 | NegLabel | 99.49 | 1.91 | 95.49 | 20.53 | 91.64 | 35.59 | 90.22 | 43.56 | 94.21 | 25.40 |
| | Ours | **99.61** | **1.54** | **96.69** | **13.82** | **92.85** | **29.69** | **93.78** | **25.78** | **95.73** | **17.71** |
| ViT-L/14 | NegLabel | 99.53 | 1.77 | 95.63 | 22.33 | 93.01 | 32.22 | 89.71 | 42.92 | 94.47 | 24.81 |
| | Ours | **99.72** | **1.21** | **96.73** | **14.88** | **93.58** | **28.41** | **92.71** | **28.16** | **95.69** | **18.17** |
| ViT-L/14-336px | NegLabel | 99.67 | 1.31 | 95.71 | 21.60 | 93.02 | 32.15 | 90.38 | 40.44 | 94.70 | 23.88 |
| | Ours | **99.79** | **0.86** | **96.82** | **14.01** | **93.64** | **27.65** | **93.12** | **27.39** | **95.84** | **17.48** |

**Analysis of different CLIP architectures.** Table 4 shows that our proposed method consistently outperforms the baseline method NegLabel by a large margin with differernt CLIP architectures, which demonstrates our effectiveness.

**Ablation of different superclass sets.** Refer to Appendix C.5 for details.

# 6   Conclusion

**Summary.** In this paper, we propose that enhancing the performance of zero-shot OOD detection theoretically requires: (1) concurrently increasing the semantic pool size and the expected activation probability of selected OOD labels; (2) ensuring low mutual dependence among the label activations. Furthermore, we analyze that the inefficacy of simply adopting larger lexicons is attributed to the introduction of numerous uncommon words and (near-)synonyms, thus failing to meet the above requirements. Observing that the original semantic pool is comprised of unmodified specific class names, we correspondingly construct a conjugated semantic pool consisting of specifically modified superclass names, each serving as a cluster center for samples sharing similar properties across different categories. Consistent with the established theory, expanding OOD label candidates with the conjugated semantic pool satisfies the requirements and achieves considerable improvements.

**Limitations and Future directions.** Our method has following limitations worth further exploration: (1) The effectiveness of CSP depends on the implicit assumption that the OOD samples exhibit a variety of distinct visual properties. When this assumption does not hold, *i.e.*, OOD samples most share similar visual properties, such as the plant images in iNaturalist, the addition of CSP results in a slight performance decline, since most newly added labels are not likely to be activated. Reducing dependency on this assumption is a valuable direction for future research. (2) In this work, we primarily focus on analyzing and optimizing the activation status of OOD labels while making no modification to the ID label set. However, there is a possibility that selecting additional labels from the semantic pool, including the CSP, to expand the ID label set could enhance the identification of difficult ID samples. We consider this a promising direction for future exploration.

## Acknowledgments and Disclosure of Funding

This work was supported in part by the National Key Research and Development Plan of China under Grant 2021ZD0112200, in part by the National Natural Science Foundation of China under Grants 62036012, U21B2044, 62236008, 62102415, U2333215, 62072286, and 62106262, in part by Beijing Natural Science Foundation under Grant 4242051.

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

# A  Proof and Derivation

The proof is partially adapted from the appendix of [29].

## A.1   Proof of Lemma 1

**Lyapunov Central Limit Theorem.** Suppose $\{s_1, ..., s_m, ...\}$ is a sequence of independent random variables, each with finite expected value $\mu_i$ and variance $\sigma_i^2$. Define $\rho_m^2 = \sum_{i=1}^m \sigma_i^2$. If for some $\delta > 0$, Lyapunov's condition

$$\lim_{m \to \infty} \frac{1}{\rho_m^{2+\delta}} \sum_{i=1}^m \mathbb{E}[|s_i - \mu_i|^{2+\delta}] = 0 \tag{10}$$

is satisfied, then a sum of $\frac{s_i - \mu_i}{\rho_m}$ converges in distribution to a standard normal random variable, as $m$ goes to infinity:

$$\frac{1}{\rho_m} \sum_{i=1}^m (s_i - \mu_i) \xrightarrow{d} \mathcal{N}(0, 1). \tag{11}$$

**Lemma 1.** *Given a sequence of independent Bernoulli random variables $\{s_1, ..., s_m\}$ with parameters $\{p_1, ..., p_m\}$, where $0 < p_i < 1$, as $m$ goes to infinity, the Poisson binomial random variable $C = \sum_{i=1}^m s_i$ converges in distribution to the normal random variable:*

$$C \xrightarrow{d} \mathcal{N}\left(\sum_{i=1}^m p_i, \sum_{i=1}^m p_i(1 - p_i)\right). \tag{12}$$

*Proof.* The proof process involves an application of the Lyapunov central limit theorem (CLT) [1], a particular form of CLT which relaxes the identical-distribution assumption. Since $s_i$ is a Bernoulli random variable with parameter $p_i$, we know that $\mu[s_i] = p_i$, $\sigma^2[s_i] = p_i(1 - p_i)$. To keep the notation uncluttered, we use $\mu_i$ and $\sigma_i^2$ for substitution. Based on the above expectation and variance values, we try to verify the Lyapunov condition: Denote $\rho_m^2 = \sum_{i=1}^m \sigma_i^2$, for some $\delta > 0$,

$$\lim_{m \to \infty} \frac{1}{\rho_m^{2+\delta}} \sum_{i=1}^m \mathbb{E}\left[|s_i - \mu_i|^{2+\delta}\right] = 0. \tag{13}$$

Analyzing the term $\mathbb{E}[|s_i - \mu_i|^{2+\delta}]$ based on $s_i$ and $\mu_i$, we know

$$\begin{aligned}
\mathbb{E}[|s_i - \mu_i|^{2+\delta}] &= (1 - \mu_i)^{2+\delta} \Pr(s_i = 1) + (0 - \mu_i)^{2+\delta} \Pr(s_i = 0) \\
&= (1 - p_i)^{2+\delta} p_i + (0 - p_i)^{2+\delta}(1 - p_i) \\
&= (1 - p_i)p_i \left((1 - p_i)^{1+\delta} + p_i^{1+\delta}\right).
\end{aligned} \tag{14}$$

Thus, we know $0 < \mathbb{E}[|s_i - \mathbb{E}[s_i]|^{2+\delta}] < 2$. Then, we analyze

$$\rho_m^{2+\delta} = \left(\sum_{i=1}^m p_i(1 - p_i)\right)^{1+\delta/2} \geq \left(\sum_{i=1}^m \varepsilon\right)^{1+\delta/2} = \varepsilon^{1+\delta/2} m^{1+\delta/2}, \tag{15}$$

where $\varepsilon = \min(p_i - p_i^2) > 0$. Based on Eqn. 14 and Eqn. 15, we have

$$0 < \frac{1}{\rho_m^{2+\delta}} \sum_{i=1}^m \mathbb{E}[|s_i - \mu_i|^{2+\delta}] \leq \frac{2m}{\varepsilon^{1+\delta/2} m^{1+\delta/2}} = \frac{2}{\varepsilon^{1+\delta/2} m^{\delta/2}}. \tag{16}$$

Thus, as $m \to \infty$, the squeeze theorem tells us that

$$\lim_{m \to \infty} \frac{1}{\rho_m^{2+\delta}} \sum_{i=1}^m \mathbb{E}[|s_i - \mu_i|^{2+\delta}] = 0. \tag{17}$$

Hence, we verify the Lyapunov condition for $C$ and $s_i$. Based on the Lyapunov CLT, we know that, as $m$ goes to infinity,

$$\frac{1}{\rho_m} \sum_{i=1}^m (s_i - \mu_i) \xrightarrow{d} \mathcal{N}(0, 1), \tag{18}$$

where $\xrightarrow{d}$ means "converges in distribution". Thus, for a sufficiently large $m$, the Poisson binomial random variable $C = \sum_{i=1}^{m} s_i$ approximately follows

$$C \sim \mathcal{N}\left(\sum_{i=1}^{m} \mu_i, \rho_m^2\right) = \mathcal{N}\left(\sum_{i=1}^{m} p_i, \sum_{i=1}^{m} p_i(1 - p_i)\right). \tag{19}$$

$\square$

## A.2 Derivation of Eq. (3)

First, we denote the cumulative distribution function (CDF) of a normal distribution with mean $\mu$ and standard deviation $\sigma$ as $\Phi(x; \mu, \sigma^2)$, which can be expressed by

$$\Phi(x; \mu, \sigma^2) = \frac{1}{2}\left[1 + \operatorname{erf}\left(\frac{x - \mu}{\sqrt{2}\sigma}\right)\right], \tag{20}$$

and its inverse function can be expressed by

$$\Phi^{-1}(x; \mu, \sigma^2) = \sqrt{2}\sigma \cdot \operatorname{erf}^{-1}(2x - 1) + \mu, \tag{21}$$

where $\operatorname{erf}(x)$ denotes the integral of the standard normal distribution from $0$ to $x$, termed as error function, and can be given by

$$\operatorname{erf}(x) = \frac{2}{\sqrt{\pi}} \int_0^x e^{-t^2} dt. \tag{22}$$

Besides, with the distributions of $C^{\text{in}}$ and $C^{\text{out}}$ given by

$$C^{\text{in}} \sim \mathcal{N}\left(mq_1, mq_1(1 - q_1) - mv_1\right), C^{\text{out}} \sim \mathcal{N}\left(mq_2, mq_2(1 - q_2) - mv_2\right), \tag{23}$$

the false positive rate (FPR) when the true positive rate (TPR) is $\lambda \in [0, 1]$, denoted by $\text{FPR}_\lambda$, can be calculated as

$$\begin{aligned} \text{FPR}_\lambda &= F_{\text{out}}(F_{\text{in}}^{-1}(\lambda)) = \Phi\left[\Phi^{-1}\left(\lambda; mq_1, mq_1(1 - q_1) - mv_1\right); mq_2, mq_2(1 - q_2) - mv_2\right] \\ &= \frac{1}{2} + \frac{1}{2} \cdot \operatorname{erf}\left(\frac{\Phi^{-1}\left(\lambda; mq_1, mq_1(1 - q_1) - Mv_1\right) - mq_2}{\sqrt{2mq_2(1 - q_2) - 2mv_2}}\right) \\ &= \frac{1}{2} + \frac{1}{2} \cdot \operatorname{erf}\left(\frac{\sqrt{2mq_1(1 - q_1) - 2mv_1}\operatorname{erf}^{-1}(2\lambda - 1) + mq_1 - mq_2}{\sqrt{2mq_2(1 - q_2) - 2mv_2}}\right) \\ &= \frac{1}{2} + \frac{1}{2} \cdot \operatorname{erf}\left(\sqrt{\frac{q_1(1 - q_1) - v_1}{q_2(1 - q_2) - v_2}}\operatorname{erf}^{-1}(2\lambda - 1) + \frac{\sqrt{m}(q_1 - q_2)}{\sqrt{2q_2(1 - q_2) - 2v_2}}\right), \end{aligned} \tag{24}$$

where $F_{\text{in}}$ and $F_{\text{out}}$ denote the cumulative distribution functions which correspond to the scores obtained by ID and OOD samples.

## A.3 Calculation process from Eq.(3) to Eq.(6)

First, from

$$\text{FPR}_\lambda = \frac{1}{2} + \frac{1}{2} \cdot \operatorname{erf}\left(\sqrt{\frac{q_1(1 - q_1) - v_1}{q_2(1 - q_2) - v_2}}\operatorname{erf}^{-1}(2\lambda - 1) + \frac{\sqrt{m}(q_1 - q_2)}{\sqrt{2q_2(1 - q_2) - 2v_2}}\right), \tag{25}$$

we know that

$$\text{FPR}_{0.5} = \frac{1}{2} + \frac{1}{2}\operatorname{erf}\left(\sqrt{\frac{m}{2}} \cdot \frac{q_0 - q_2 + u(r|q_0, q_2)}{\sqrt{q_2(1 - q_2) - v_2}}\right). \tag{26}$$

Denote $z = \sqrt{\frac{m}{2}} \cdot \frac{q_0 - q_2 + u(r|q_0, q_2)}{\sqrt{q_2(1-q_2) - v_2}}$, then we can derive that

$$
\begin{aligned}
G(r|q_0, q_2, u, M) &= \frac{\partial \text{FPR}_{0.5}}{\partial r} = \frac{\partial \text{FPR}_{0.5}}{\partial z} \frac{\partial z}{\partial m} \frac{\partial m}{\partial r} = \frac{1}{2} \frac{\partial \text{erf}(z)}{\partial z} \frac{\partial z}{\partial m} \frac{\partial m}{\partial r} \\
&= \frac{M}{2} \cdot \frac{2 e^{-z^2}}{\sqrt{\pi}} \cdot \frac{\frac{1}{2\sqrt{m}}(q_0 - q_2 + u) + \frac{\sqrt{m}}{M} \frac{\partial u}{\partial r}}{\sqrt{2 q_2(1-q_2) - 2v_2}} \\
&= \frac{M e^{-z^2}}{2\sqrt{2\pi}} \cdot \frac{q_0 - q_2 + u + \frac{2m}{M} \frac{\partial u}{\partial r}}{\sqrt{m} \sqrt{q_2(1-q_2) - v_2}}.
\end{aligned}
\tag{27}
$$

Denote $w = \frac{1}{\sqrt{m}} \left( q_0 - q_2 + u + \frac{2m}{M} \frac{\partial u}{\partial r} \right)$, then we have

$$
\begin{aligned}
\frac{\partial w}{\partial r} &= \frac{M}{m} \left( \left( \frac{1}{M} \frac{\partial u}{\partial r} + \frac{2}{M} \frac{\partial u}{\partial r} + \frac{2m}{M^2} \frac{\partial^2 u}{\partial r^2} \right) \sqrt{m} - \frac{1}{2\sqrt{m}} \left( q_0 - q_2 + u + \frac{2m}{M} \frac{\partial u}{\partial r} \right) \right) \\
&= \frac{M}{2m^{\frac{3}{2}}} \left( \frac{4m^2}{M^2} \frac{\partial^2 u}{\partial r^2} + \frac{4m}{M} \frac{\partial u}{\partial r} - q_0 + q_2 - u \right) \\
&\geq \frac{M}{2m^{\frac{3}{2}}} \left( -\frac{4m^2}{M^2} \frac{\partial u}{\partial r} + \frac{4m}{M} \frac{\partial u}{\partial r} - q_0 + q_2 - u \right) \\
&= \frac{M}{2m^{\frac{3}{2}}} \left( 4r(1-r) \frac{\partial u}{\partial r} + (q_2 - q_0 - u) \right) \geq 0,
\end{aligned}
\tag{28}
$$

where the derivation from the second to the third line utilizes an assumption on the limited range of the curvature of the function $u(r)$, aka $|u''| \leq u'$. While recognizing the challenge of consistently maintaining this assumption in complex real-world scenarios, we consider it reasonable for use in intuitive quantitative analyses to simplify derivations. Therefore, we come to the conclusion that $G'(r) = \frac{\partial^2 \text{FPR}_{0.5}}{\partial r^2} \geq 0$. Besides, according to Eqn. 27, we have

$$
\lim_{r \to 0^+} G(r) = \lim_{r \to 0^+} \frac{\sqrt{M} e^{-z^2}}{2\sqrt{2\pi}} \cdot \frac{q_0 - q_2}{\sqrt{r} \sqrt{q_2(1-q_2) - v_2}} = \lim_{r \to 0^+} \frac{\kappa(q_0 - q_2)}{2\sqrt{r}} = -\infty, \tag{29}
$$

$$
\lim_{r \to 1} G(r) = \frac{e^{-z^2}}{\sqrt{2\pi}} \cdot \frac{\sqrt{M} u'(r=1|q_0, q_2)}{\sqrt{q_2(1-q_2) - v_2}} = \kappa u'(r=1) \geq 0, \tag{30}
$$

where $\kappa = (M/2\pi)^{\frac{1}{2}} (q_2(1-q_2) - v_2)^{-\frac{1}{2}} e^{-z^2} > 0$.

### A.4 Analysis of some variables in Eq.(8)

Defined as the lower bound of $q_1$, $q_0$ represents the expected probability of the OOD label, which is most dissimilar to the ID label space, being activated by ID images. When the semantic pool size is large enough, this OOD label has very different meaning from ID labels, thus its expected probability $q_0$ of being activated by ID images is close to zero. With the assumption that the function $u(r)$ is linear, from Eqn. 7 we can derive that $r_0 = 1/3$, which is a constant unrelated to other factors. Empirical evidence also indicates that the change of $r_0$ is relatively slight. Besides, $v_2$ is defined as the variance of the probabilities of OOD labels being activated by OOD samples. When the semantic pool is large enough, this variance changes very slightly with further expansion. Therefore, we come to the conclusion that variables $q_0$, $r_0$, and $v_2$ remain nearly constant with a sufficiently large semantic pool, thus exerting marginal impact to the right side of Eqn. 8.

### A.5 Analysis of a condition in Eq.(9)

By adjusting its shape parameters, $\alpha$ and $\beta$, the Beta distribution can flexibly simulate a range of distinct probability distribution profiles. This flexibility is particularly useful in statistical modeling and analysis where the behavior of probabilities needs to be accurately described. Assuming that $p_i^{\text{out}}$ follows a Beta distribution, we know that

$$
q_2 = \mathbb{E}_i[p_i^{\text{out}}] = \frac{\alpha}{\alpha + \beta}, \quad v_2 = \text{Var}_i[p_i^{\text{out}}] = \frac{\alpha\beta}{(\alpha + \beta)^2 (\alpha + \beta + 1)}. \tag{31}
$$

If $q_2 - 2v_2 \geq 0$ does not hold, it means that

$$\frac{\alpha}{\alpha + \beta} - \frac{2\alpha\beta}{(\alpha + \beta)^2(\alpha + \beta + 1)} < 0, \tag{32}$$

which is equivalent to

$$\alpha^2 + 2\alpha\beta + \beta^2 + \alpha - \beta < 0, \tag{33}$$

and $\alpha < \min\{0.5, \beta\} < 1$ is a necessary condition for the above equation. In this situation, the Beta distribution exhibits a pronounced peak at 0 and a long, thin tail stretching towards 1. In our experiments, we observe that the similarity distribution between OOD input images and OOD text labels exhibits a distinct unimodal concentration, with probability densities near 0 and 1 approaching zero, which is entirely different from the distribution shape derived theoretically. Consequently, we can conclude that in the vast majority of practical cases, the following inequality holds,

$$q_2 + q_0 - 2q_0q_2 - 2v_2 = q_2 - 2v_2 + q_0(1 - 2q_2) \geq 0. \tag{34}$$

## B  Datasets and Lexicons

### B.1  Main benchmark

We mainly evaluate our method on the widely-used ImageNet-1k OOD detection benchmark [26]. This benchmark utilizes the large-scale ImageNet-1k dataset as the ID data, and select samples from iNaturalist [60], SUN [66], Places [73], and Textures [8] as the OOD data. The categories of the OOD data have been manually selected to prevent overlap with the classes of ImageNet-1k.

**ImageNet-1k**, also referred to as ILSVRC 2012, is a subset of the larger ImageNet dataset [9]. This dataset encompasses 1,000 object classes and includes 1,281,167 images for training, 50,000 images for validation, and 100,000 images for testing. In the widely used benchmark for OOD detection organized by [26], the validation set of ImageNet-1k is designated as the ID data.

**iNaturalist** [60] is a fine-grained dataset containing 859,000 images across more than 5,000 species of plants and animals. [26] randomly sample 10,000 images from 110 manually selected plant classes which are not present in ImageNet-1k as the OOD data.

**SUN** [66] is a scene database which includes 130,519 images from 397 categories. [26] sample 10,000 images from 50 nature-related classes that do not overlap with ImageNet-1k as OOD data.

**Places** [73] is another scene dataset containing more than 2.5 million images covering more than 205 scene categories with more than 5,000 images per category. [26] manually select 50 categories from this dataset and then randomly sample 10,000 images as OOD data.

**Textures** [8], also referred to as Describable Textures Dataset (DTD), consists of 5,640 images from 47 categories of textural patterns inspired from human perception. There are 120 images for each category. [26] use the entire dataset as OOD data.

### B.2  Datesets of hard OOD detection

We also evaluate our method on the hard OOD detection tasks as shown in Table 2. Specifically, the ID-vs-OOD dataset pairs includes ImageNet-10 vs ImageNet-20, ImageNet-10 vs ImageNet-100, ImageNet-1k vs ImageNet-O [22], WaterBirds [50]-vs-Placebg, etc.

**ImageNet-O** [22] is a dataset of adversarially filtered examples for ImageNet OOD detectors. To create this dataset, the authors delete examples of ImageNet-1k from ImageNet-22k, and then select examples that a ResNet-50 [19] model misclassifies as belonging to an ImageNet-1k class with high confidence. This dataset contains 2,000 images across 200 classes. In a hard OOD detection task, we use ImageNet-1k as the ID data and use ImageNet-O as OOD data.

**WaterBirds** [50] is constructed by combining bird photographs from the CUB-200 dataset [61] with image backgrounds from the Places dataset [73]. Therefore, WaterBirds keeps the same size with CUB-200, *i.e.*, it contains 11,788 images from 200 bird classes. To construct this dataset, the authors label each bird as waterbird or landbird and place it on one image of water background or land background. In a hard OOD detection task, we use WaterBirds as the ID data and use its background images as OOD data.

## B.3 Datasets of various ID data

As depicted in Table 7, our method is evaluated against baseline methods, including MCM [40] and NegLabel [29], across various ID datasets. These datasets encompass (1) specialized domain-focused datasets such as Stanford-Cars [31], CUB-200 [61], Oxford-Pet [45], and Food-101 [2]; (2) subsets of ImageNet, including ImageNet10, ImageNet20, and ImageNet100; (3) ImageNet domain shift datasets, namely ImageNet-Sketch [62], ImageNet-A [22], ImageNet-R [20], and ImageNetV2 [48].

**Stanford-Cars** [31] contains 16,185 images of 196 classes of cars. Classes are typically at the level of Make, Model, Year, *e.g.*, 2012 Tesla Model S or 2012 BMW M3 coupe. The data is split into 8,144 training images and 8,041 testing images.

**CUB-200** [61], formally recognized as Caltech-UCSD Birds-200-2011, comprises 11,788 images across 200 bird subcategories, with 5,994 images for training and 5,794 for testing.

**Oxford-Pet** [45] is a 37 category pet dataset with roughly 200 images for each class created by the Visual Geometry Group at Oxford. The total image number is 7,390.

**Food-101** [2] dataset consists of 101 food categories with 750 training and 250 test images per category, making a total of 101k images.

**ImageNet-Sketch** [62] dataset consists of 50,889 images, approximately 50 images for each of the 1,000 ImageNet classes. The dataset was created using Google Image searches for "sketch of {Class Name}", specifically limiting results to the "black and white" color scheme.

**ImageNet-A** [22] is a dataset of 7,500 real-world adversarially filtered images, which are misclassified by a ResNet-50 ImageNet classifier, from 200 classes. The user-tagged images are downloaded from websites including iNaturalist, Flicker, and DuckDuckGo.

**ImageNet-R** [20], formally recognized as ImageNet-Renditions, contains 30,000 images of ImageNet objects from 200 classes with different textures and styles.

**ImageNetV2** [48] contains 10,000 new images across the 1,000 categories of ImageNet-1k. The new images are gathered from the same source of ImageNet to avoid bias.

## B.4 Lexicons

As shown in Table 9, we conduct experiments with lexicons of different sizes, and observe that simply adopting larger lexicons does not yield consistent performance improvement. From each lexicon, we select all the nouns to construct the original semantic pool, and use all the adjectives to construct the additional conjugated semantic pool for expansion.

**WordNet** [39] is a large lexical database of English. Nouns, verbs, adjectives and adverbs are grouped into sets of cognitive synonyms, each expressing a distinct concept. In our experiments, we use the 70K nouns and adjectives to construct a semantic pool.

**Common-20K**[3] is a list of the 20,000 most common English words in order of frequency, as determined by n-gram frequency analysis of the Google's Trillion Word Corpus. In our experiments, we use the 17K nouns and adjectives to construct a semantic pool.

**Part-of-Speech Tagging**[4] is a 370K English words corpus. In our experiments, we use the 319K nouns and adjectives to construct a semantic pool.

# C   More Results and Analysis

## C.1   Validation of consistency between methodology and theoretical framework

Consistent with our established theory, expanding label candidates with the CSP satisfies the requirements derived in Section 3.1: (1) Concurrently enlarging the semantic pool size $M$ and the expected activation probability $q_2$ of OOD labels; (2) Ensuring that there is no severe mutual dependence among the activations of selected OOD labels.

---

[3] https://github.com/first20hours/google-10000-english
[4] https://www.kaggle.com/datasets/ruchi798/part-of-speech-tagging

(1) The enlargement of the semantic pool size $M$ is evident. Besides, since the superclasses used in constructing the CSP typically include broad semantic objects, the property clusters encompass samples from numerous potential OOD categories. Therefore, their centers have much higher expected probabilities of being activated by OOD samples, which brings an increase in $q_2$. In Table 5, we present the expected softmax scores for a single OOD label from both the original semantic pool and the CSP. These scores, averaged across OOD samples, serve as an approximation of $q_2$, which is defined as the expected probability of OOD labels being activated by OOD samples. Table 5 reveals that the average score of our CSP across four OOD datasets is distinctly higher than that of the original pool, indicating that this expansion leads to an increase in $q_2$.

Table 5: The expected Softmax scores of a single OOD label, an approximation of $q_2$, from the original semantic pool and the conjugated semantic pool, scaled up by a factor of 1000.

| Semantic Pools | The expected Softmax score of a single OOD label | | | | Average |
| | iNaturalist | SUN | Places | Textures | |
|---|---|---|---|---|---|
| Original / Conjugated | **0.1356** / 0.0308 | 0.0923 / **0.2176** | 0.0864 / **0.2213** | 0.0404 / **0.4435** | 0.0887 / **0.2283** |

The effectiveness of the CSP is based on the implicit assumption that OOD samples exhibit various visual properties. However, the degree of visual diversity varies across different OOD datasets, resulting in different expected probabilities of OOD labels in the CSP being activated, as reflected in the varying scores of conjugated labels shown in Table 5. For instance, plant images in iNaturalist have limited visual diversity, leading to low scores for conjugated labels, whereas the Texture dataset, with its higher visual diversity, exhibits the opposite phenomenon. We can observe that across different OOD datasets, there is a correlation between these scores and the performance improvements achieved by our method: the score is lower on iNaturalist compared to the original pool, relatively higher on SUN and Places, and significantly higher on Textures. Consequently, our method achieves only modest gains on iNaturalist, normal improvements on SUN and Places, and substantial enhancements on Textures. This correlation further corroborates the validity of our theory.

(2) Since the labels in CSP are centers of property clusters, while the labels in the original semantic pool are centers of category clusters, it is highly improbable that numerous synonym pairs would exist between these two semantic pools. Our statistical analysis supports this claim: we calculate the average maximum similarity between each label and other labels within the semantic pool, a metric which reflects the proportion of synonymous pairs within the pool and tends to increase monotonically as the semantic pool expands. Our findings indicate that only $3.94\%$ of the original labels find more similar counterparts in the expanded CSP, resulting in a negligible increase in the aforementioned metric from $0.8721$ to $0.8726$. As a result, the mutual dependence between the new and original labels is relatively low, and the functions of labels from the CSP will not be overshadowed, enhancing the likelihood that an OOD image locates close to an OOD cluster center.

## C.2   Random analysis

Table 6 shows the results of our method under random seeds from 0 to 9, whose average is reported in the main text. It is evident that the performance of our method is minimally impacted by randomness, consistently exhibiting superior efficacy.

## C.3   Various ID datasets

As depicted in Table 7, our method is evaluated against baseline methods, including MCM [40] and NegLabel [29], across various ID datasets. These datasets encompass (1) specialized domain-focused datasets such as Stanford-Cars [31], CUB-200 [61], Oxford-Pet [45], and Food-101 [2]; (2) subsets of ImageNet, including ImageNet10, ImageNet20, and ImageNet100; (3) ImageNet domain shift datasets, namely ImageNet-Sketch [62], ImageNet-A [22], ImageNet-R [20], and ImageNetV2 [48]. Our proposed method consistently achieves satisfactory results on all the above ID datasets. For example, our method outperforms NegLabel by 9.23% and 11.71% evaluated by FPR95 with ImageNet-A and ImageNet-R as the ID data, respectively.

Table 6: Mean and standard deviation of OOD detection performance across various random seeds with CLIP-B/16 on ImageNet-1k as ID data. Performance metrics are presented as percentages.

| Seeds | iNaturalist | | SUN | | Places | | Textures | | Average | |
| --- | --- | --- | --- | --- | --- | --- | --- | --- | --- | --- |
| | AUROC↑ | FPR95↓ | AUROC↑ | FPR95↓ | AUROC↑ | FPR95↓ | AUROC↑ | FPR95↓ | AUROC↑ | FPR95↓ |
| 0 | 99.61 | 1.54 | 96.69 | 13.82 | 92.85 | 29.69 | 93.78 | 25.78 | 95.73 | 17.71 |
| 1 | 99.61 | 1.51 | 96.67 | 13.57 | 92.99 | 29.03 | 93.69 | 25.57 | 95.74 | 17.42 |
| 2 | 99.60 | 1.56 | 96.68 | 13.61 | 92.95 | 29.18 | 93.74 | 26.10 | 95.74 | 17.61 |
| 3 | 99.60 | 1.56 | 96.69 | 13.57 | 92.93 | 29.21 | 93.64 | 26.01 | 95.72 | 17.59 |
| 4 | 99.60 | 1.54 | 96.65 | 13.65 | 92.87 | 29.26 | 94.10 | 24.95 | 95.81 | 17.35 |
| 5 | 99.60 | 1.55 | 96.66 | 13.61 | 92.88 | 29.37 | 94.01 | 24.79 | 95.79 | 17.33 |
| 6 | 99.60 | 1.53 | 96.70 | 13.53 | 92.92 | 29.27 | 93.87 | 25.96 | 95.77 | 17.57 |
| 7 | 99.61 | 1.54 | 96.61 | 13.78 | 92.85 | 29.50 | 93.88 | 25.74 | 95.74 | 17.64 |
| 8 | 99.60 | 1.53 | 96.64 | 13.90 | 92.89 | 29.52 | 94.09 | 24.75 | 95.81 | 17.43 |
| 9 | 99.60 | 1.56 | 96.64 | 13.58 | 92.88 | 29.20 | 93.80 | 25.59 | 95.73 | 17.48 |
| Mean | 99.60 | 1.54 | 96.66 | 13.66 | 92.90 | 29.32 | 93.86 | 25.52 | 95.76 | 17.51 |
| Std | 0.00 | 0.02 | 0.03 | 0.12 | 0.04 | 0.19 | 0.15 | 0.48 | 0.03 | 0.12 |

Table 7: OOD detection performance comparison on various ID datasets.

| ID datasets | Methods | iNaturalist | | SUN | | Places | | Textures | | Average | |
| --- | --- | --- | --- | --- | --- | --- | --- | --- | --- | --- |
| | | AUROC↑ | FPR95↓ | AUROC↑ | FPR95↓ | AUROC↑ | FPR95↓ | AUROC↑ | FPR95↓ | AUROC↑ | FPR95↓ |
| Stanford-Cars | MCM | 99.77 | 0.05 | 99.95 | 0.02 | 99.89 | 0.24 | 99.96 | 0.02 | 99.89 | 0.08 |
| | NegLabel | 99.99 | 0.01 | 99.99 | 0.01 | 99.99 | 0.03 | 99.99 | 0.01 | 99.99 | 0.02 |
| | Ours | **100.00** | **0.00** | **100.00** | **0.00** | **99.99** | **0.02** | **100.00** | **0.00** | **100.00** | **0.01** |
| CUB-200 | MCM | 98.24 | 9.83 | 99.10 | 4.93 | 98.57 | 6.65 | 98.75 | 6.97 | 98.67 | 7.10 |
| | NegLabel | **99.96** | 0.18 | **99.99** | 0.02 | **99.90** | 0.33 | 99.99 | 0.01 | **99.96** | **0.14** |
| | Ours | **99.96** | **0.16** | **99.99** | 0.03 | 99.88 | 0.37 | **100.00** | **0.00** | **99.96** | **0.14** |
| Oxford-Pet | MCM | 99.38 | 2.85 | 99.73 | 1.06 | 99.56 | 2.11 | 99.81 | 0.80 | 99.62 | 1.71 |
| | NegLabel | 99.99 | 0.01 | 99.99 | 0.02 | **99.96** | **0.17** | 99.97 | 0.11 | **99.98** | **0.08** |
| | Ours | **100.00** | **0.00** | **100.00** | **0.00** | **99.96** | 0.21 | 99.97 | 0.14 | **99.98** | 0.09 |
| Food-101 | MCM | 99.78 | 0.64 | 99.75 | 0.90 | 99.58 | 1.86 | 98.62 | 4.04 | 99.43 | 1.86 |
| | NegLabel | 99.99 | 0.01 | 99.99 | 0.01 | **99.99** | **0.01** | 99.60 | 1.61 | 99.89 | 0.41 |
| | Ours | **100.00** | **0.00** | **100.00** | **0.00** | **99.99** | **0.01** | **99.63** | **1.40** | **99.91** | **0.35** |
| ImageNet10 | MCM | 99.80 | 0.12 | 99.79 | 0.29 | 99.62 | 0.88 | 99.90 | 0.04 | 99.78 | 0.33 |
| | NegLabel | 99.83 | **0.02** | **99.88** | 0.20 | **99.75** | 0.71 | **99.94** | 0.02 | **99.85** | 0.24 |
| | Ours | **99.84** | 0.04 | **99.88** | **0.13** | 99.74 | **0.61** | **99.94** | **0.02** | **99.85** | **0.20** |
| ImageNet20 | MCM | 99.66 | 1.02 | 99.50 | 2.55 | 99.11 | 4.40 | 99.03 | 2.43 | 99.33 | 2.60 |
| | NegLabel | 99.95 | 0.15 | 99.51 | 1.93 | 98.97 | 4.40 | 99.11 | 2.41 | 99.39 | 2.22 |
| | Ours | **99.96** | **0.10** | **99.65** | **1.07** | **99.13** | **3.20** | **99.24** | **1.68** | **99.50** | **1.51** |
| ImageNet100 | MCM | 96.77 | 18.13 | 94.54 | 36.45 | 94.36 | 34.52 | 92.25 | 41.22 | 94.48 | 32.58 |
| | NegLabel | 99.87 | 0.57 | 97.89 | 11.26 | 96.25 | 19.15 | 96.00 | 20.37 | 97.50 | 12.84 |
| | Ours | **99.90** | **0.46** | **98.78** | **4.84** | **97.19** | **13.31** | **98.16** | **8.83** | **98.51** | **6.86** |
| ImageNet-Sketch | MCM | 87.74 | 63.06 | 85.35 | 67.24 | 81.19 | 70.64 | 74.77 | 79.59 | 82.26 | 70.13 |
| | NegLabel | 99.34 | 2.24 | 94.93 | 22.73 | 90.78 | 38.62 | 89.29 | 46.10 | 93.59 | 27.42 |
| | Ours | **99.49** | **1.60** | **96.41** | **15.30** | **92.51** | **31.41** | **92.95** | **29.86** | **95.34** | **19.54** |
| ImageNet-A | MCM | 79.50 | 76.85 | 76.19 | 79.78 | 70.95 | 80.51 | 61.98 | 86.37 | 72.16 | 80.88 |
| | NegLabel | 98.80 | 4.09 | 89.83 | 44.38 | 82.88 | 60.10 | 80.25 | 64.34 | 87.94 | 43.23 |
| | Ours | **99.15** | **2.91** | **91.06** | **42.70** | **85.16** | **59.87** | **93.08** | **30.50** | **92.11** | **34.00** |
| ImageNet-R | MCM | 83.22 | 71.51 | 80.31 | 74.98 | 75.53 | 76.67 | 67.66 | 83.72 | 76.68 | 76.72 |
| | NegLabel | 99.58 | 1.60 | 96.03 | 15.77 | 91.97 | 29.48 | 90.60 | 35.67 | 94.55 | 20.63 |
| | Ours | **99.79** | **0.89** | **98.49** | **6.16** | **95.41** | **18.46** | **96.44** | **10.16** | **97.53** | **8.92** |
| ImageNetV2 | MCM | 91.79 | 45.90 | 89.88 | 50.73 | 86.52 | 56.25 | 81.51 | 69.57 | 87.43 | 55.61 |
| | NegLabel | 99.40 | 2.47 | 94.46 | 25.69 | 90.00 | 42.03 | 88.46 | 48.90 | 93.08 | 29.77 |
| | Ours | **99.54** | **1.76** | **96.10** | **17.16** | **91.66** | **34.12** | **92.76** | **29.65** | **95.02** | **20.67** |

## C.4 Empirical evidence supporting our assertions

**Performance trends related to the ratio $r$.** In Fig. 1 and Table 8, we present the FPR95 performances of our method and NegLabel against a progressively increasing ratio $r$, which represents the proportion of selected OOD labels in the whole semantic pool. The color gradations displayed in the table clearly illustrate an initial improvement in model performance followed by a subsequent decline as the ratio $r$ increases. This trend aligns with our derivation in Section 3.1.

**Effect of simple lexicon expansion.** In Fig. 2 and Table 9, we assess whether adopting larger lexicons enhances performances. Our findings indicate that it does not always hold. When the semantic pool covers the vast majority of common words, further expansion will introduce an excessive number of uncommon words and (near-)synonyms, thus failing to meet the derived requirements for theoretical

performance enhancement. The inefficacy of simple lexicon expansion indicates that viable expansion manners move beyond merely selecting words from existing lexicons.

Table 8: OOD detection performance evaluated by the FPR95 metric with different candidate selection ratios $r$. The results of our method and the baseline method NegLabel share similar trends.

| Ratio $r$ | FPR95 Performance of Ours | | | | | FPR95 Performance of NegLabel | | | | |
|---|---|---|---|---|---|---|---|---|---|---|
| | iNaturalist | SUN | Places | Textures | Average | iNaturalist | SUN | Places | Textures | Average |
| 0.02 | 1.28 | 23.50 | 37.34 | 26.10 | 22.06 | 1.31 | 33.66 | 46.26 | 50.32 | 32.89 |
| 0.05 | 1.17 | 19.07 | 33.91 | 23.42 | 19.39 | 1.26 | 25.90 | 39.81 | 44.96 | 27.98 |
| 0.10 | 1.28 | 15.44 | 30.78 | 25.41 | 18.23 | 1.54 | 22.20 | 36.80 | 43.40 | 25.99 |
| 0.15 | 1.54 | 13.82 | 29.69 | 25.78 | 17.71 | 1.95 | 20.84 | 36.00 | 43.40 | 25.55 |
| 0.20 | 1.82 | 13.60 | 29.61 | 25.43 | 17.62 | 2.46 | 20.70 | 36.29 | 43.49 | 25.74 |
| 0.25 | 2.17 | 13.75 | 29.70 | 25.55 | 17.79 | 2.93 | 21.42 | 37.05 | 44.04 | 26.36 |
| 0.30 | 2.39 | 13.70 | 29.67 | 25.94 | 17.93 | 3.30 | 22.04 | 37.71 | 45.00 | 27.01 |
| 0.40 | 2.96 | 14.27 | 30.34 | 27.32 | 18.72 | 4.22 | 22.48 | 38.91 | 46.95 | 28.14 |
| 0.50 | 3.58 | 14.90 | 30.96 | 28.69 | 19.53 | 5.01 | 23.16 | 39.76 | 48.44 | 29.09 |
| 0.60 | 4.18 | 15.46 | 31.33 | 29.43 | 20.10 | 5.79 | 24.17 | 40.69 | 50.30 | 30.24 |
| 0.80 | 5.19 | 16.28 | 31.77 | 32.70 | 21.49 | 7.67 | 25.75 | 41.72 | 53.85 | 32.25 |
| 1.00 | 6.27 | 17.41 | 32.60 | 35.00 | 22.82 | 9.26 | 27.25 | 42.85 | 56.29 | 33.91 |

Table 9: Evaluation with different corpus sources. "Size" refers to the size of semantic pools.

| Source | Size | Method | OOD Datasets | | | | | | | | Average | |
|---|---|---|---|---|---|---|---|---|---|---|---|---|
| | | | iNaturalist | | SUN | | Places | | Textures | | | |
| | | | AUROC | FPR95 | AUROC | FPR95 | AUROC | FPR95 | AUROC | FPR95 | AUROC | FPR95 |
| Commom-20K | 17K | NegLabel | 86.91 | 65.43 | 95.03 | 24.22 | 91.52 | 34.83 | 83.69 | 67.75 | 90.50 | 43.02 |
| | | Ours | 90.50 | 47.94 | 95.89 | 19.13 | 92.35 | 30.92 | 87.70 | 51.52 | 92.06 | 36.56 |
| WordNet-v2.0 | 70K | NegLabel | 99.49 | 1.91 | 95.49 | 20.53 | 91.64 | 35.59 | 90.22 | 43.56 | 94.21 | 25.40 |
| | | Ours | 99.61 | 1.54 | 96.69 | 13.82 | 92.85 | 29.69 | 93.78 | 38.49 | 95.73 | 17.71 |
| WordNet-v3.0 | 121K | NegLabel | 99.44 | 2.18 | 94.73 | 25.12 | 90.50 | 41.85 | 89.46 | 47.59 | 93.53 | 29.19 |
| | | Ours | 99.65 | 1.37 | 96.43 | 15.73 | 92.25 | 31.62 | 93.69 | 26.26 | 95.51 | 18.75 |
| Part-of-Speech | 319K | NegLabel | 98.57 | 6.23 | 94.21 | 26.50 | 89.95 | 44.56 | 88.09 | 51.19 | 92.71 | 32.12 |
| | | Ours | 98.82 | 5.02 | 95.14 | 21.46 | 90.99 | 38.49 | 91.90 | 32.70 | 94.21 | 24.42 |

## C.5 Analysis of superclasses in conjugated semantic pool

The indices 1 through 14 in Table 10 represent the following superclasses, listed in alphabetical order: *area, creature, environment, item, landscape, object, pattern, place, scene, space, structure, thing, view, vista*. Table 10 displays the outcomes of multiple runs with 4, 7, and 10 randomly selected superclasses. Although the selection of different superclasses results in some performance fluctuations, any selection significantly enhances performance compared to not employing CSP (as shown in the first row of the table), and achieves state-of-the-art results.

Acute readers may be concerned about performance fluctuations caused by different superclass sets. However, despite the specific OOD categories being unknown in real-world applications, it is likely that an approximate range of OOD superclasses can be estimated in advance based on the deployment scenario and empirical evidence. Generally, users can preset a suitable superclass set to achieve satisfactory performance enhancements provided by the conjugated semantic pool.

# D   Visualization

In this section, we present visualization results of images picked from the ImageNet-1k OOD detection benchmark. Each subfigure includes the original image, the ground-truth label (for ID images only), the image name in the dataset, and the top-5 softmax scores for ID labels (orange), OOD labels from the original semantic pool (green), and OOD labels from the conjugated semantic pool (blue).

## D.1   In-distribution examples

In Figs. 4 and 5, we present ID examples that have been correctly classified into the ground-truth ID class with high and low confidence, respectively. Fig. 6 presents ID examples that have been correctly classified into the ID class group but assigned the wrong specific classes. Figs. 7 and 8 display failure cases where the ID image is misclassified into labels of the original semantic pool or our conjugated semantic pool, respectively.

Table 10: Analysis of the number of the superclasses constructing the conjugated Semantic Pool.

| Superclasses of Conjugated Semantic Pool | | | | | | | | | | | | | | iNaturalist | | SUN | | Places | | Textures | | Average | |
|---|---|---|---|---|---|---|---|---|---|---|---|---|---|---|---|---|---|---|---|---|---|---|---|
| 1 | 2 | 3 | 4 | 5 | 6 | 7 | 8 | 9 | 10 | 11 | 12 | 13 | 14 | AUROC | FPR95 | AUROC | FPR95 | AUROC | FPR95 | AUROC | FPR95 | AUROC | FPR95 |
| | | | | | | | | | | | | | | 99.63 | 1.40 | 96.25 | 16.63 | 92.02 | 33.90 | 86.91 | 57.62 | 93.70 | 27.39 |
| | ✓ | | | | | ✓ | | | | | | ✓ | ✓ | 99.62 | 1.48 | 96.68 | 13.25 | 93.08 | 27.92 | 89.87 | 45.73 | 94.81 | 22.10 |
| ✓ | | | | | | | | | ✓ | ✓ | | | ✓ | 99.61 | 1.54 | 96.64 | 13.88 | 92.98 | 28.91 | 90.77 | 43.05 | 95.00 | 21.85 |
| ✓ | | ✓ | | | | | | | ✓ | | | ✓ | | 99.62 | 1.51 | 96.84 | 12.87 | 93.05 | 28.48 | 89.91 | 45.09 | 94.86 | 21.99 |
| | | ✓ | | | | | ✓ | ✓ | | | | | ✓ | 99.57 | 1.64 | 96.29 | 15.44 | 92.38 | 31.96 | 95.12 | 20.59 | 95.84 | 17.41 |
| | | | | | ✓ | ✓ | | | | ✓ | | | | 99.61 | 1.48 | 97.13 | 12.22 | 93.52 | 27.07 | 90.80 | 43.53 | 95.27 | 21.08 |
| Average of above 5 runs using 4 superclasses | | | | | | | | | | | | | | 99.61 | 1.53 | 96.72 | 13.53 | 93.00 | 28.87 | 91.29 | 39.60 | 95.15 | 20.88 |
| ✓ | ✓ | | ✓ | | ✓ | ✓ | | | | | | ✓ | ✓ | 99.59 | 1.6 | 96.2 | 15.42 | 92.32 | 31.95 | 94.67 | 22.06 | 95.70 | 17.76 |
| ✓ | ✓ | | ✓ | ✓ | | | | | ✓ | ✓ | | | ✓ | 99.62 | 1.5 | 96.88 | 12.97 | 93.25 | 27.63 | 90.57 | 42.87 | 95.08 | 21.24 |
| ✓ | | ✓ | ✓ | | | | | | ✓ | ✓ | ✓ | ✓ | | 99.62 | 1.52 | 96.53 | 14.22 | 92.71 | 30.09 | 91.12 | 41.21 | 95.00 | 21.76 |
| | | ✓ | | | | ✓ | ✓ | | ✓ | ✓ | | | ✓ | 99.59 | 1.59 | 96.28 | 15.45 | 92.44 | 31.84 | 94.67 | 22.66 | 95.75 | 17.89 |
| | ✓ | | | ✓ | ✓ | ✓ | | | ✓ | ✓ | | | | 99.61 | 1.52 | 96.78 | 13.47 | 93.09 | 28.51 | 91.20 | 40.74 | 95.17 | 21.06 |
| Average of above 5 runs using 7 superclasses | | | | | | | | | | | | | | 99.61 | 1.55 | 96.53 | 14.31 | 92.76 | 30.00 | 92.45 | 33.91 | 95.34 | 19.94 |
| ✓ | ✓ | ✓ | ✓ | ✓ | ✓ | ✓ | ✓ | | | | | ✓ | ✓ | 99.60 | 1.56 | 96.72 | 13.66 | 92.83 | 29.63 | 94.36 | 23.85 | 95.88 | 17.18 |
| ✓ | ✓ | | ✓ | ✓ | | | ✓ | ✓ | ✓ | ✓ | | ✓ | ✓ | 99.62 | 1.49 | 96.93 | 12.36 | 93.28 | 27.19 | 90.1 | 45.07 | 94.98 | 21.53 |
| ✓ | ✓ | ✓ | | | ✓ | ✓ | | | ✓ | | | ✓ | | 99.60 | 1.57 | 96.00 | 16.16 | 92.04 | 33.29 | 94.63 | 22.59 | 95.57 | 18.40 |
| ✓ | | ✓ | | ✓ | ✓ | ✓ | | ✓ | ✓ | ✓ | | | ✓ | 99.61 | 1.55 | 96.61 | 13.79 | 92.85 | 29.89 | 94.22 | 24.13 | 95.82 | 17.34 |
| | ✓ | ✓ | ✓ | ✓ | ✓ | | ✓ | ✓ | ✓ | ✓ | ✓ | | | 99.61 | 1.51 | 96.81 | 13.15 | 93.14 | 28.41 | 90.55 | 43.17 | 95.03 | 21.56 |
| Average of above 5 runs using 10 superclasses | | | | | | | | | | | | | | 99.61 | 1.54 | 96.61 | 13.82 | 92.83 | 29.68 | 92.77 | 31.76 | 95.46 | 19.20 |
| ✓ | ✓ | ✓ | ✓ | ✓ | ✓ | ✓ | ✓ | ✓ | ✓ | ✓ | ✓ | ✓ | ✓ | 99.61 | 1.54 | 96.69 | 13.82 | 92.85 | 29.69 | 93.78 | 25.78 | 95.73 | 17.71 |

We observe that, for ID samples, incorrect OOD detection results mainly stem from the following reasons: (1) Low image clarity or VLM limitations: Due to the low clarity of the images or the limited capabilities of the VLM, the VLM provides incorrect classification results. For instance, the category of image *ImageNet_ILSVRC2012_val_00002364* in Fig. 7 is *"coho (silver salmon)"*, but CLIP mistakenly classifies it as *"chum salmon"*, thereby identifying it as an OOD sample. (2) Inaccurate ground truth labels: Some images have ground truth labels that are not precise enough. For example, the image *ImageNet_ILSVRC2012_val_00004471* in Fig. 8 is labeled as *"coil or spiral"*, which is not commonly used to refer to a spiral staircase. This leads the model to classify the image as a *"helter-skelter structure"*, a more accurate OOD category. (3) Multiple elements in images: Certain images contain multiple elements that correspond to several appropriate labels. Although we selected OOD labels with low similarity to the ID label space, it does not ensure that OOD labels are entirely unrelated to the specific ID images. For instance, the image *ImageNet_ILSVRC2012_val_00003031* in Fig. 8 is labeled as *"doormat"*, but *"cursive view"* and *"frosty view"* are also suitable descriptions, resulting in incorrect OOD detection.

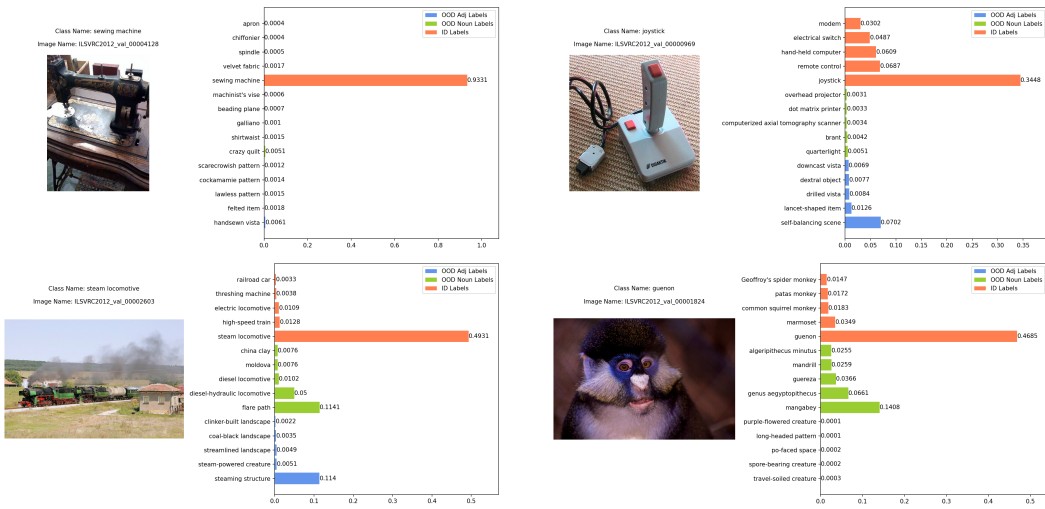

Figure 4: ID Examples of correct OOD detection, correct classification, and high confidence.

## D.2 Out-of-distribution examples

Figs. 9,11,13, and 15 show OOD images from the iNaturalist, Places, SUN, and Textures datasets, respectively, that have been correctly classified as OOD samples. Contrarily, Figs. 10,12,14, and 16 display the failure cases, where OOD images are misclassified as ID ones.

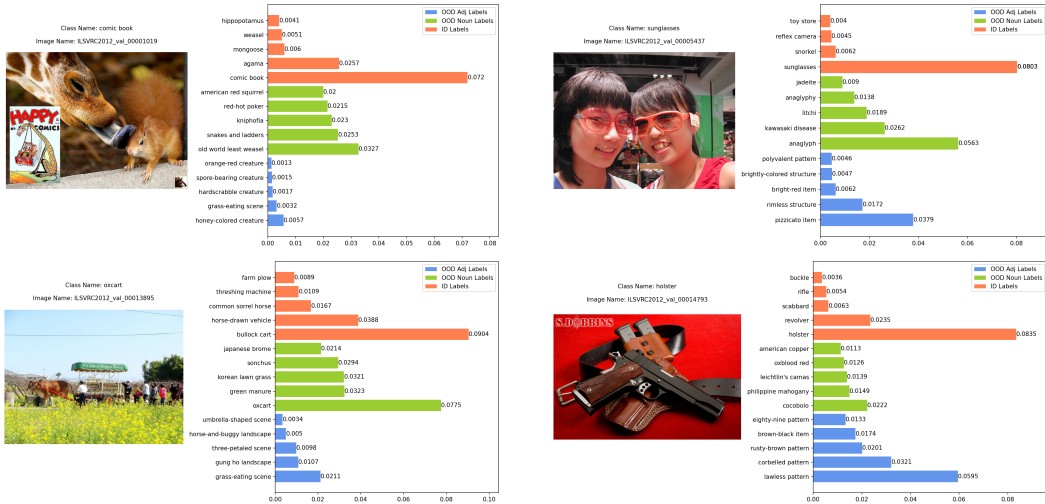

Figure 5: ID Examples of correct OOD detection, correct classification, and low confidence.

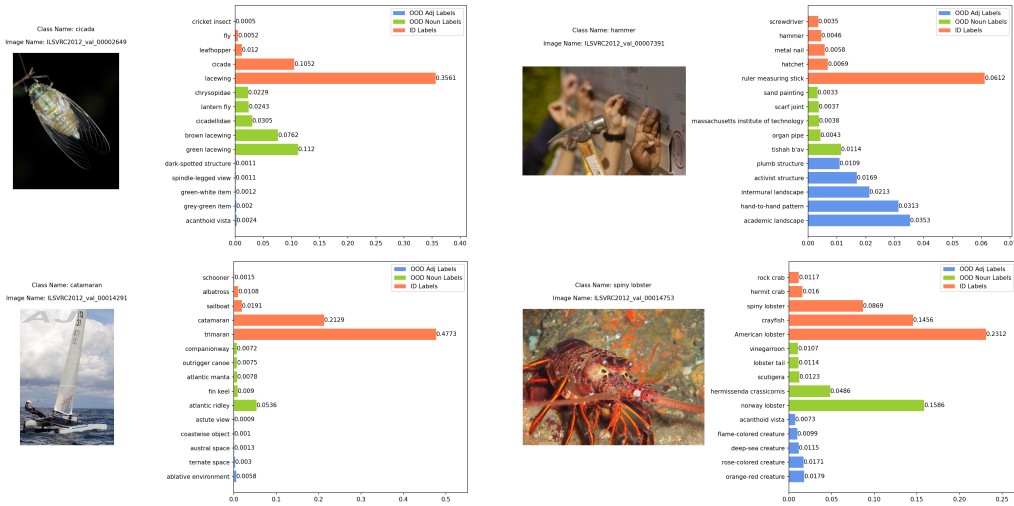

Figure 6: ID Examples of correct OOD detection and incorrect classification.

We observe that there are two primary reasons for OOD misdetection results in OOD datasets: (1) Absence of suitable OOD labels: In the process of selecting potential OOD labels, elements in the semantic pool that have a high similarity to ID labels are discarded. This may lead to the absence of corresponding labels for OOD images. For example, in Fig. 16, the image named *waffled_0103* depicts a waffle. However, the OOD label candidates do not include the label *"waffle"*, resulting in the image being incorrectly classified as the ID category *"waffle iron"*. (2) Presence of ID category objects within some OOD images: For instance, in Fig. 12, the image *s_ski_slope_00004560* from the Places dataset, whose label is *"ski slope"*, depicts a man skiing on a ski slope. Actually, classifying it as the ID category *"ski"* is entirely correct, and this image should not be considered an OOD sample. Similarly, in Fig. 16, the label of the image *striped_0032* from the Texture dataset, which shows part of a zebra, is *"striped"*, but it is also reasonable that CLIP directly classifies it as the ID category *"zebra"*. Thus, although the ImageNet-1k OOD detection benchmark established by [26] has been widely used, constructing more accurate and comprehensive OOD datasets remains crucial for further advancements in this field. We will pursue this as a future research direction.

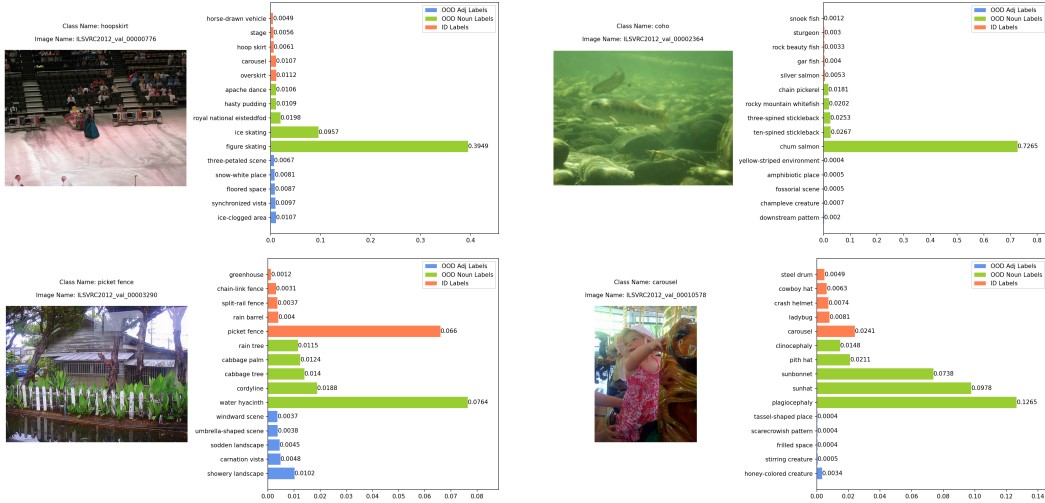

Figure 7: ID Examples of incorrect OOD detection classified into the **original** semantic pool.

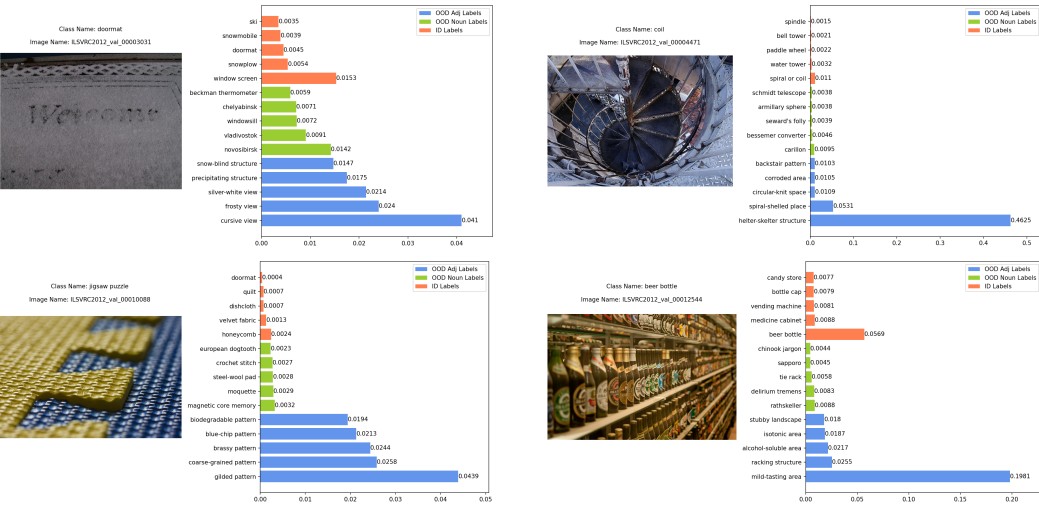

Figure 8: ID Examples of incorrect OOD detection classified into the **conjugated** semantic pool.

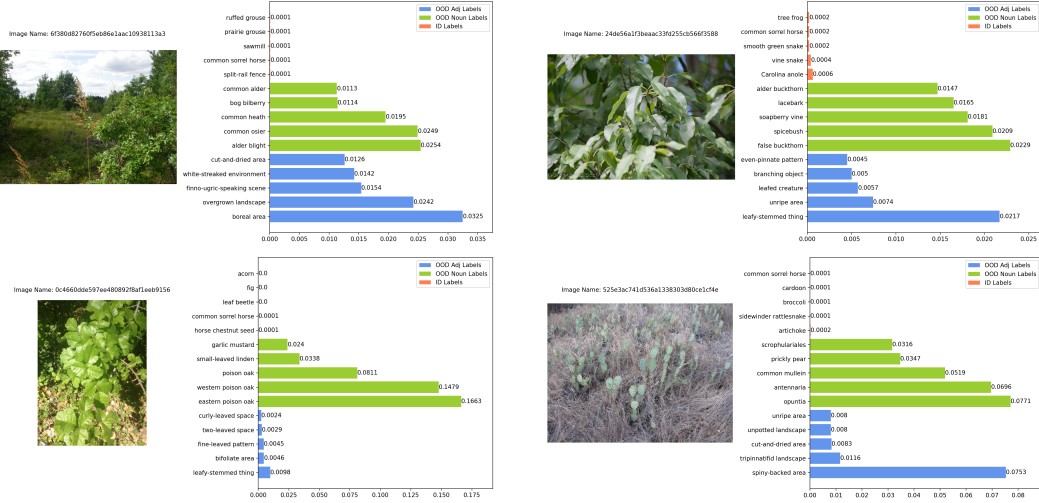

Figure 9: OOD Examples of correct OOD detection from **iNaturalist**.

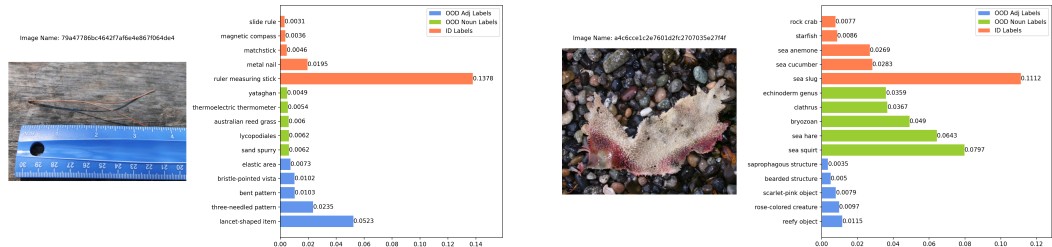

Figure 10: OOD Examples of incorrect OOD detection from **iNaturalist**.

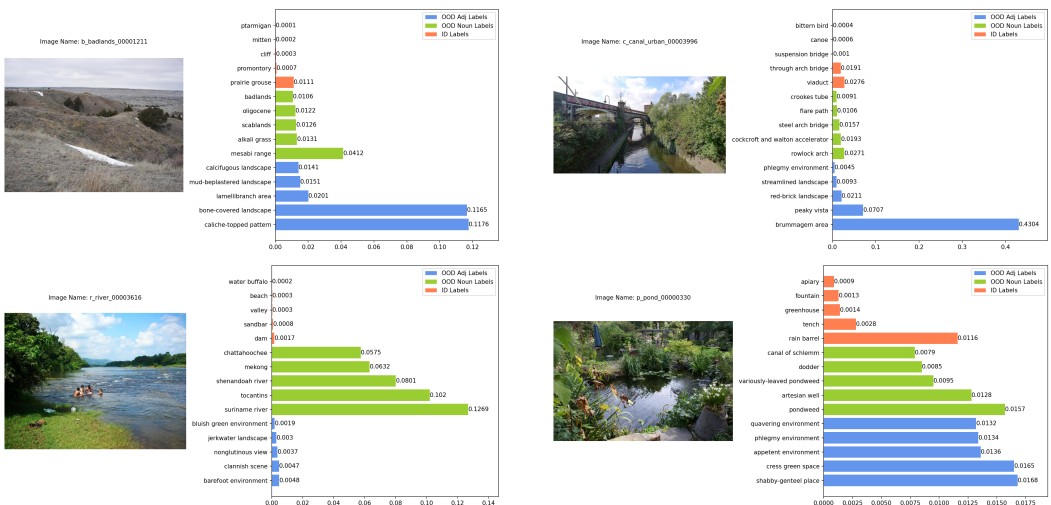

Figure 11: OOD Examples of correct OOD detection from **Places**.

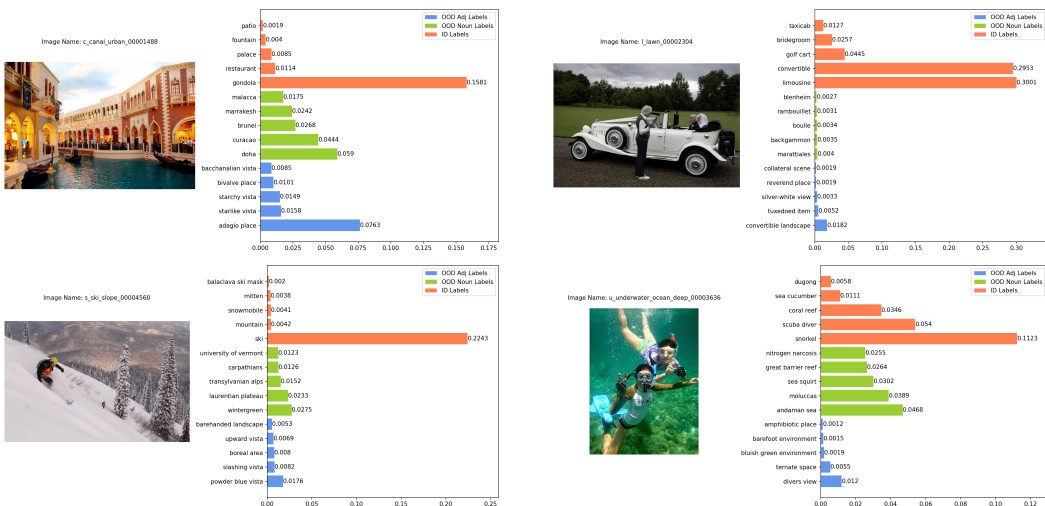

Figure 12: OOD Examples of incorrect OOD detection from **Places**.

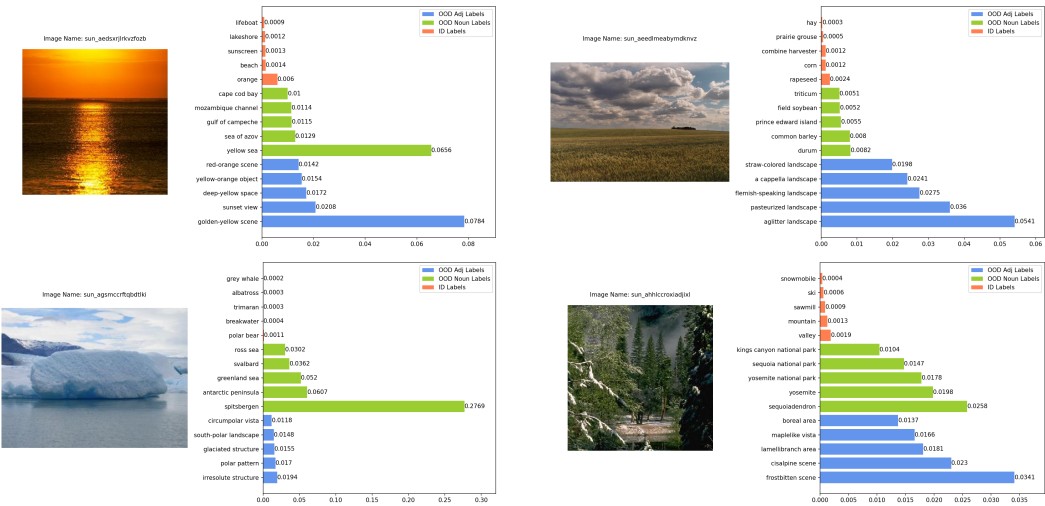

Figure 13: OOD Examples of correct OOD detection from **SUN**.

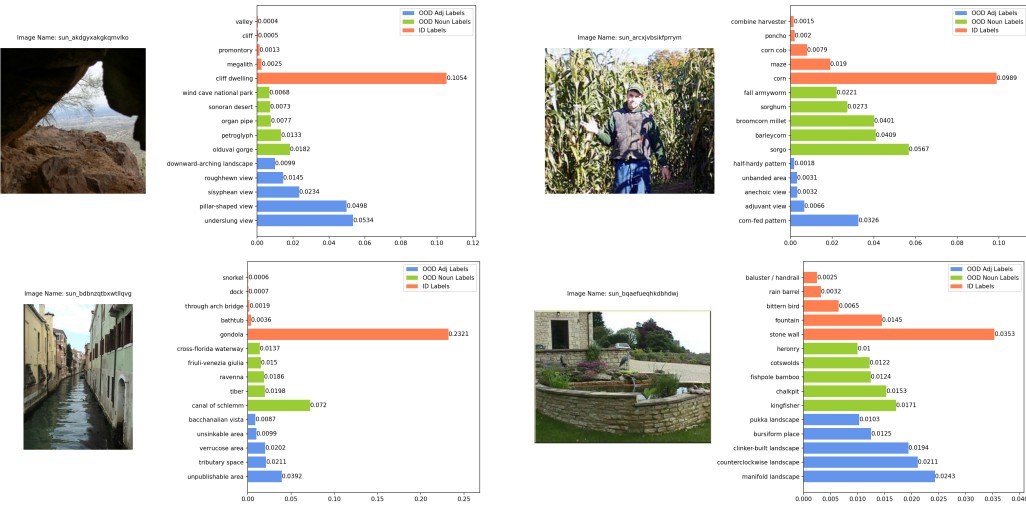

Figure 14: OOD Examples of incorrect OOD detection from **SUN**.

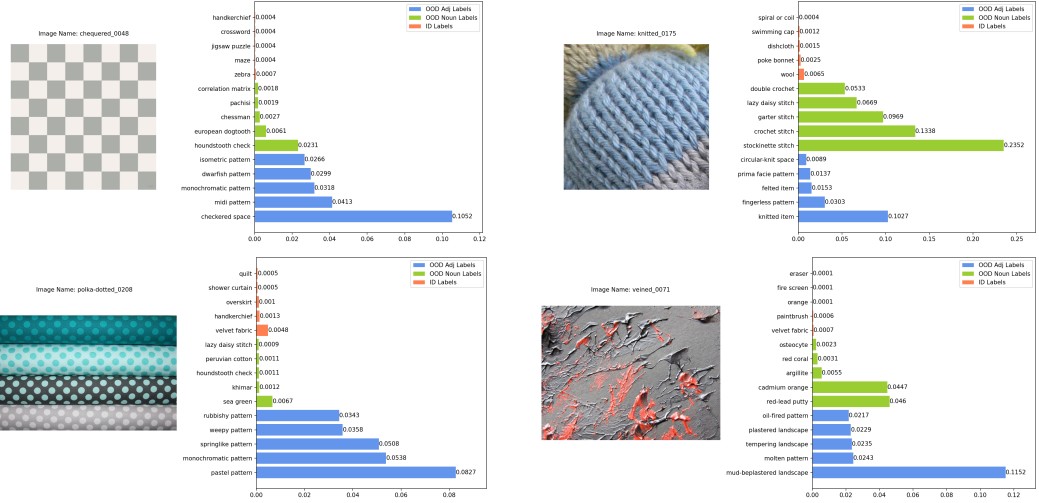

Figure 15: OOD Examples of correct OOD detection from **Textures**.

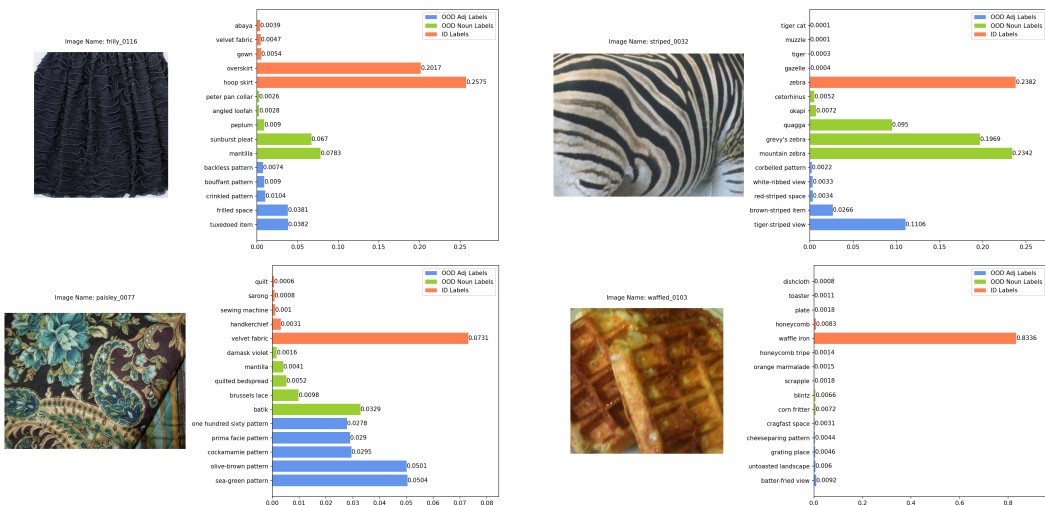

Figure 16: OOD Examples of incorrect OOD detection from **Textures**.

