# OpenReview forum: "Conjugated Semantic Pool Improves OOD Detection with Pre-trained Vision-Language Models"
_NeurIPS.cc/2024/Conference — NeurIPS 2024 poster_

### Official Review · Reviewer_KrTH · 2024-06-29

**Soundness:** 4
**Presentation:** 4
**Contribution:** 4
**Rating:** 7
**Confidence:** 5

**Summary:**

This paper proposes a novel method utilizing pre-trained vision-language models (VLMs) to enhance out-of-distribution (OOD) detection. Their core idea lies in constructing a Conjugated Semantic Pool (CSP) to enrich the pool from which OOD labels are drawn. Unlike simply expanding lexicons with synonyms and uncommon words, the CSP leverages modified superclass names that serve as cluster centers for samples with similar properties across categories. This, the authors theorize, not only increases pool size but also amplifies the activation probability of OOD labels while maintaining low label dependence. Extensive experiments demonstrate significant performance improvements over existing approaches.

**Strengths:**

Originality: The paper introduces a unique concept, conjugated semantic labels, for enhancing the semantic pool, offering a valuable contribution to OOD detection.
Theoretical Underpinnings: The authors provide a robust theoretical framework, including mathematical models and justifications demonstrating how the CSP improves OOD detection.
Empirical Validation: Extensive experiments showcase the effectiveness of the proposed method, achieving significant advancements over current state-of-the-art techniques.
Generalizability: The method's applicability across diverse VLM architectures suggests broad utility and adaptability.
Clear Problem Definition: The paper clearly identifies limitations in existing methods, particularly regarding simple lexicon expansion, and proposes a well-reasoned solution.

**Weaknesses:**

1. Ablation Study & OOD Score Function:
The ablation study in Table 11 showcases that competitive performance on iNaturalist and SUN datasets is achievable without the CSP. However, it remains unclear how the proposed OOD score function (designed by the authors) contributes to performance when CSP is not employed. Further clarification is needed on: (1)Performance Source: If strong results are achievable without CSP on these datasets, what factors besides the CSP are driving the overall performance improvement observed in the main experiments? (2) OOD Score Function Usage: How can the authors' OOD score function be effectively utilized in scenarios where CSP is not implemented?
2. Synonym Handling and CSP Overlap:
The paper acknowledges the limitations of including synonyms in larger lexicons. However, it lacks a deeper exploration of how the CSP specifically addresses this issue. Additionally, the potential for semantic overlap within the CSP itself needs to be addressed. It would be beneficial to see a discussion on: (1) CSP vs. Synonyms: How does the design of the CSP inherently mitigate the problems introduced by synonyms in traditional lexicon expansion? (2) Mitigating Overlap: How do the authors handle potential semantic overlap between elements within the CSP? Are there strategies to ensure distinctiveness between labels?
3. Systematic Analysis on Place and Texture Datasets:
A more systematic analysis of the CSP's impact on the Place and Texture datasets, particularly concerning the design principles of semantic pooling, would be valuable. The authors' own experiments suggest that designing effective semantic pooling requires considering multiple factors. Therefore, a more in-depth exploration of: (1) Place and Texture Performance: How does the CSP specifically improve performance on Place and Texture datasets? Can these improvements be linked to specific design choices within the CSP? (2) Semantic Pooling Design Factors: Building on the authors' findings, what factors are crucial for designing effective semantic pooling mechanisms, especially in the context of Place and Texture datasets?

Overall, this paper proposes a promising approach to improve OOD detection by innovatively using conjugate semantic labels to expand the semantic pool, which is supported by theoretical analysis and empirical results. However, I still have some questions about the method that the authors need to address.

**Questions:**

See Weakness

**Limitations:**

See Weakness

---

> ### Author Rebuttal · Authors · 2024-08-05
>
> Thank you for your constructive comments and the positive rating! In the following, your comments are concisely mentioned (due to the length limitation) and then followed by our point-by-point responses.
>
> > 1.  Ablation Study & OOD Score Function: ...
>
> Thank you for the insightful comment! The OOD score function we adopted is proposed by NegLabel (ICLR2024) without modifications. The performance improvement on certain datasets when not using CSP stems from minor adjustments we made to the lexicon used to construct the original semantic pool. Specifically:
>
> 1.  **Performance Source:** In the baseline presented in Table 11, we removed all adjectives and proper nouns, including personal names, organization names, and quantifiers, from the semantic pool constructed by NegLabel. This led to a higher expected activation probability of negative labels for OOD images in the iNaturalist and SUN datasets. However, this did not result in an overall performance improvement. Compared to the NegLabel method, the performance gains on the iNaturalist and SUN datasets were offset by performance drops on the Places and Texture datasets, resulting in a decrease in overall model performance (25.40% -> 27.39% on FPR95).
>
> 2.  **OOD Score Function Usage:** Our Conjugated Semantic Pool (CSP) is a method for expanding the semantic pool. Without using CSP, we still have the original semantic pool composed of simple noun labels, which can be used with the OOD score function. We will include additional details in the implementation part of the revised version to ensure clearer explanations.
>
> > 2.  Synonym Handling and CSP Overlap: ...
>
> Thank you for the valuable comment! We provided related discussion in Appendix D.1 (L1143-1152). Specifically:
>
> 1.  **CSP vs. Synonyms:** When the size of a traditional lexicon reaches a certain point, further expansion inevitably introduces a large number of synonyms, which are unlikely to contribute to performance improvement. However, using CSP to expand the semantic pool effectively mitigates this issue without introducing numerous synonyms, since the labels in CSP are centers of property clusters, while the labels in the original semantic pool are centers of category clusters. As discussed in Appendix D.1, our statistical analysis supports this claim: we calculate the average maximum similarity between each label and other labels within the semantic pool—a metric that reflects the proportion of synonymous pairs within the pool and tends to increase monotonically as the semantic pool expands. Our findings indicate that only 3.94% of the original labels find more similar counterparts in the expanded CSP, resulting in a negligible increase in the aforementioned metric from 0.8721 to 0.8726. Consequently, the mutual dependence between the new and original labels is relatively low.
>
> 3.  **Mitigating Overlap in CSP:** When constructing the conjugated semantic pool, each label has a different adjective and a randomly selected superclass from a set of 14 superclasses. For two labels in the CSP, significant semantic overlap would only occur if both their adjectives and superclass words are synonymous, thus the likelihood of such overlap within the CSP is very low.
>
>
> Of course, we acknowledge that CSP, as a method for expanding the semantic pool, cannot eliminate synonyms that already exist within the original pool. However, the main limitation of synonyms is that their activations are highly dependent on each other, making it difficult for synonym-based expansion to improve performance in line with the theoretical guidance of mathematical models. The presence of a small number of synonyms should not significantly harm model performance.
>
> > 3.  Systematic Analysis on Place and Texture Datasets: ...
>
> Thanks for the valuable comment! We provide a more in-depth exploration of these issues and will add them to the revised version:
>
> 1. **Place and Texture Performance:** We provide related discussion in Appendix D.1 (L1132-1142) and provide supporting data in Table 5, which may be helpful to clarify this issue. Specifically, by linking the performance improvements on the Places and Textures datasets (as well as SUN) to the design choices of CSP, we conclude that because the CSP was constructed using a diverse range of adjectives and superclass words with broad semantic meanings, labels in the CSP can be considered as centers of property clusters. Therefore, the effectiveness of the CSP is based on the implicit assumption that OOD samples exhibit various visual properties. The images in SUN, Places, and Texture primarily depict natural environments and textures, which have strong visual property diversity, leading to relatively larger performance improvements. In contrast, iNaturalist, where images are mostly focused on various plants with limited visual property diversity, does not benefit from the inclusion of the CSP.
>
> 2.  **Semantic Pooling Design Factors:** Focusing on the following factors of CSP can contribute to performance improvement: (1) When constructing the CSP, avoid reusing adjectives and instead pair each adjective with a randomly selected superclass to minimize semantic overlap between labels. (2) When setting up the superclass set, aim to include a broad semantic range in the set. Generally, more superclasses tend to bring better performance due to increased diversity. (3) As mentioned in our response to Reviewer gvS3’s comment 3, the ratio of negative label selection impacts performance, and selecting negative labels from the CSP achieves the best performance at 40%.
>
> **In the end, thanks again for all your time and consideration in reviewing our paper!**

---

> > ### Comment · Reviewer_KrTH · 2024-08-08
> > **Comment**
> >
> > Thanks for your response. It has solved my concerns. And I have raised my rating.

---

### Official Review · Reviewer_p3ao · 2024-07-08

**Soundness:** 3
**Presentation:** 3
**Contribution:** 2
**Rating:** 6
**Confidence:** 5

**Summary:**

This paper presents a novel zero-shot out-of-distribution (OOD) detection pipeline that enhances performance by expanding the semantic pool with a conjugated semantic pool (CSP), which consists of modified superclass names that cluster similar properties from different categories. The approach aims to increase the activation probability of selected OOD labels by OOD samples while ensuring low mutual dependence among these labels. By moving beyond traditional lexicon-based expansion and using the CSP, the paper's method outperforms existing works by 7.89% in FPR95, demonstrating a significant improvement in OOD detection without the need for additional training data.

**Strengths:**

1. The paper is well-written and easy to understand.
2. The supplement to the shortcomings of the NegLabel theory is very enlightening.
3. Extensive experiments demonstrate the superiority of the method.

**Weaknesses:**

1. The primary contribution of this work is its enhancement of the semantic pool expansion issue within NegLabel, representing an incremental step within the NegLabel framework.

2. There is ambiguity regarding the distinction between citations and contributions. For instance, the content related to Lemma 1 in both the preliminary section and the appendix A.1, A.2 seems to originate directly from the original NegLabel paper, yet this source is not explicitly referenced in the main text and appendix.

3. In Section 3.3, the authors think that OOD images may not include their true label "white peacock" among the negative labels is a issue. However, this argument appears debatable. As evidenced in NegLabel's visualizations and Figures 6 and 7 of this paper, OOD images may not consistently exhibit the highest similarity to their ground truth (GT) label and could show significant similarity with multiple negative labels. Thus, even if the true GT label of an OOD image is absent from the negative labels, the NegLabel method can still effectively detect OOD data.

4. From an implementation standpoint, how does including terms like "smartphone" and "cellphone" as negative labels impact the overall OOD score? The claim in Section 3.2 about the impact of synonym disruption on Lemma 1's independence seems somewhat overstated.

5. I am curious whether, aside from the semantic pool, all other aspects of the technical implementation directly derive from NegLabel. Specifically, does the NegMining Algorithm, the OOD score calculation, and the Grouping Strategy align exactly with those in NegLabel? This aspect requires further clarification.

6. Additionally, is Figure 3 a representation of actual distribution distance data or a schematic created by the authors? Utilizing real experimental data to support the findings presented in Figure 3 would strengthen its validity. If it is a schematic, it may introduce subjective biases from the authors.

7. The paper extensively discusses the limitations of previous methods and the intended outcomes in Section 3, yet provides relatively minimal detail on the actual implementation of the proposed method. It would be beneficial for the authors to expand on the specifics of their proposed approach in Section 3.3. For instance, further elaboration on how superclass selection is conducted and integrated with adjectives would be valuable.

8. According to the description in the paper, the method used to select cluster centers appears crucial. Have the authors considered employing clustering techniques for superclass selection rather than relying solely on manual selection?

9. Another point of confusion arises regarding how variations in corpus size impact the acquisition of semantic pools. Does this necessitate selecting additional superclasses?

**Questions:**

see Cons

**Limitations:**

Yes

---

> ### Author Rebuttal · Authors · 2024-08-05
>
> Thanks for your constructive comments and positive rating! Below, we briefly restate your comments (due to length limitations) and then provide our point-by-point responses.
>
> > 1. Enhancement of semantic pool expansion within NegLabel
>
> Thanks for the valuable comment. Our method builds on the NegLabel framework, introducing further innovations. **NegLabel is the most effective paradigm for this task.** In addressing its limitations, we proposed a performance modeling approach that better aligns with experimental results and identified the conditions for effective semantic pool expansion. This allowed us to optimize this step, resulting in satisfactory performance improvements. Therefore, we believe our work **provides meaningful theoretical and performance advancements, showcasing unique innovation**.
>
> > 2.  Ambiguity between citations and contributions.
>
> We originally structured the paper to clearly differentiate contributions, with NegLabel's contributions discussed in the Preliminary section and our own in the Methodology. We apologize for any ambiguity and will clarify this in the revised version.
>
> In Preliminary, we have cited NegLabel at **L99 and L104**, to introduce NegLabel's methodology (L97-103) and its theoretical performance modeling (L104-124). We **will make these references more explicit** and add the statement in Appendix A.1 and A.2: “**The proof in this part is adapted from the appendix of [24]**”.
>
> Additionally, we have cited NegLabel [24] a total of **16** times, with "NegLabel" appearing **20** times, throughout the paper. This should make it clear that our method is based on NegLabel, upon which we have introduced further innovations.
>
> > 3.  Debatable augument about not including true labels.
>
> Thank you for the insightful comment! We agree that NegLabel can effectively detect OOD data when the true label (GT) of the OOD image is excluded from the negative labels. However, this isn’t always guaranteed. If an OOD image’s most similar classes are not OOD classes, the model may fail to detect it as OOD.
>
> For instance, in a scenario where the ID classes are "*hognose snake*", "**basset hound**", and "**Afghan hound**", and the OOD classes are "**toy terrier**", "*green snake*", and "*king snake*" (all are ImageNet-1k categories), if "**toy terrier**" is not included in negative labels, OOD detection for "toy terrier" images might fail, even with a strong zero-shot classifier, because **their most similar classes are within the ID set**.
>
> Moreover, even if NegLabel works when the GT is excluded from negative labels, **it doesn't mean including the GT in negative labels is pointless**. Ideally, with a sufficiently powerful VLM, OOD images should exhibit the highest similarity to their GT labels. Including the GT in negative labels **can naturally result in higher OOD scores for OOD images**, improving detection in more challenging scenarios.
>
> > 4.  Impact of synonym disruption on Lemma 1's independence.
>
> Thanks for the valuable comment! Regarding this issue, we suggest considering **an extreme case**: if we duplicate each negative label 10 times, will the model's performance improve simply because the semantic pool is larger? Clearly not, and this contradicts the derivation in NegLabel and our work, which advocates for larger semantic pools. The reason is that this duplication violates the independence assumption in Lemma 1. When there is significant interdependence among Bernoulli variables, actual performance can deviate substantially from the mathematical model. **Adding synonyms has a similar effect to duplicating labels.**
>
> However, we do **NOT** consider that using Lemma 1 in NegLabel and our work for performance modeling is flawed, as any mathematical modeling is an approximation of real-world conditions under ideal assumptions.
>
> > 5.  Question about technical implementations from NegLabel.
>
> Yes, we adopted the exact same NegMining algorithm, OOD score calculation, and Grouping Strategy as in NegLabel. In the revised version, we will clarify these technical details more explicitly.
>
> > 6.  Question about Figure 3.
>
> Fig 3 is indeed a schematic diagram created by us to illustrate our motivations. Following your suggestion, we plan to generate visualizations using real experimental data to support the findings presented in Fig 3 in the revised version.
>
> > 7.  More implementation details.
>
> We will incorporate more specific implementation details in Sec 3.3 and 5.1 of the paper. Specifically, we manually chose terms with broad semantic meanings, such as "item", "place", "creature", as superclasses. Experiments showed that various superclass sets can all achieve satisfactory performance improvements. The combination of adjectives with superclasses is entirely random: for each different adjective, we randomly select a superclass from the set and pair it with the adjective to form a phrase.
>
> > 8.  Employing clustering techniques for superclass selection.
>
> Thanks for the constructive comment. The clustering technique is not employed for superclass selection for its **higher complexity and weaker interpretability**. Clustering results vary with factors like the algorithm, sample size, and number of clusters, adding complexity. In contrast, our method introduces no additional hyperparameters, requires no extra data or training, and enhances interpretability by using explicit superclass terms instead of abstract cluster centers.
>
> > 9.  How variations in corpus size impact semantic pools.
>
> Regardless of the corpus size, our method for constructing CSP is consistent, as mentioned in response to comment 7, and the pseudocode is provided in the rebuttal PDF file. Therefore, the size of the CSP is corresponds to the number of adjectives in the corpus, but a larger corpus does not necessarily require more superclasses.
>
> **In the end, we express our sincerest gratitude for your time and thoughtful consideration!**

---

> > ### Comment · Reviewer_p3ao · 2024-08-08
> >
> > Thanks for your responses and most of my concerns are addressed. I have raised my score.

---

### Official Review · Reviewer_gvS3 · 2024-07-12

**Soundness:** 3
**Presentation:** 3
**Contribution:** 4
**Rating:** 6
**Confidence:** 5

**Summary:**

This paper explores how to set potential OOD labels to facilitate OOD detection with vision-language models. The paper first conducts a theoretical analysis, revealing that in addition to increasing the negative label space, it is also important to increase the probability of negative labels being activated and to reduce mutual dependence between negative labels. Therefore, the paper proposes a new strategy for constructing negative labels by introducing modified superclass names to construct a conjugated semantic pool (CSP). Experiments are conducted on standard benchmarks.

**Strengths:**

1. The paper theoretically analyzes the criteria for selecting negative labels and proposes a new method for designing negative labels.
2. The motivation is clear, and the method design is straightforward and effective.
3. The proposed method achieves state-of-the-art results on standard benchmarks.

**Weaknesses:**

While the overall method design is agreeable, there are some details that need clarification:

1. In Line 276, the authors claim that CSPs overlapping with ID classes will not be selected as negative labels. How is this implemented, and does it significantly impact the results?
2. In Line 321, the authors introduce the superclasses used and mention combining these superclasses with adjectives to form the CSP. Could you provide more details on the adjectives used and the selection process? Is the pairing with superclasses done randomly?
3. How many negative labels are ultimately used? Beyond the 10,000 classes in the negative label pool, how many new negative labels are introduced? Does the number of newly introduced negative labels significantly affect OOD detection performance?
4. In my experiments, on ImageNet, performance initially improves and then declines as the number of negative labels increases; however, in experiments with CIFAR as the ID dataset, performance continuously improves with more negative labels. Have the authors observed similar phenomena, and can they provide an explanation?
5. It is suggested to follow Neglabel and refer to the introduced classes as "negative labels" instead of "OOD labels" to avoid confusion with the labels of practical OOD datasets.

**Questions:**

See weaknesses

---

> ### Author Rebuttal · Authors · 2024-08-05
>
> We express our sincere gratitude for your valuable comments. Below, we first reiterate your comments, subsequently providing our detailed responses to each point.
>
> > 1.  In Line 276, the authors claim that CSPs overlapping with ID classes will not be selected as negative labels. How is this implemented, and does it significantly impact the results?
>
> Thank you for the insightful comment! In the revised version, we will clarify the statement in L276 to avoid any ambiguity.
>
> Apart from the negative label selection strategy based on inverse similarity provided by NegLabel, **we do NOT have additional implementation to ensure that CSP labels with semantic overlap with ID classes are not selected**. What we intended to convey in L276 is that the labels in the CSP are also selected as negative labels **based on inverse similarity**, thereby reducing the likelihood of CSPs with semantic overlap with ID classes being chosen as negative labels. There are no specific strategies in our method that can completely prevent semantic overlap between CSP and ID classes.
>
> > 2.  In Line 321, the authors introduce the superclasses used and mention combining these superclasses with adjectives to form the CSP. Could you provide more details on the adjectives used and the selection process? Is the pairing with superclasses done randomly?
>
> Thank you for your suggestion!
>
> When constructing the conjugated semantic pool, the adjectives we used were unfiltered adjectives from the lexicon adopted ("adj.all.txt" in WordNet). For selecting negative labels, we employed the NegMining algorithm provided by NegLabel. The pseudocode is provided in the rebuttal PDF file.
>
> The pairing of superclasses and adjectives was entirely random; that is, for each adjective, we randomly selected a superclass from the set introduced in L321 to form a phrase.
>
> > 3.  How many negative labels are ultimately used? Beyond the 10,000 classes in the negative label pool, how many new negative labels are introduced? Does the number of newly introduced negative labels significantly affect OOD detection performance?
>
> Thank you for your question!
>
> **We ultimately used 8,492 negative labels, of which 7,005 were simple noun labels from NegLabel, and 1,487 were from our constructed CSP.** The proportion of negative labels selected from the semantic pool was 15%, consistent with NegLabel. However, the total number of noun labels we obtained was smaller than in NegLabel because we removed word categories such as personal names, organization names, and quantifiers, which generally have little utility in activating OOD images. All the ablation experiments were conducted after removing these categories.
>
> The number of newly introduced negative labels has a noticeable impact on OOD detection performance. When keeping the selection proportion of noun labels at 15%, and gradually adjusting the proportion of negative labels selected from CSP from 2% (198 labels) to 100% (9,916 labels), the model’s performance (FPR95) increased from **23.22%**, peaked at **16.66%** when the proportion was 40% (3,966 labels), and then began to decline, reaching **18.64%** when the proportion was 100%. **The specific experimental results are presented in Table 1 of the Rebuttal PDF file.** In other words, if a separate selection ratio parameter for CSP was set, a higher performance than reported could be achieved at 40%. However, to avoid parameter tuning, we directly adopted the 15% ratio parameter adopted by NegLabel.
>
> > 4.  In my experiments with CIFAR as the ID dataset, performance continuously improves with more negative labels. Have the authors observed similar phenomena, and can they provide an explanation?
>
> Thank you for your comment!
>
> We had not previously conducted experiments using CIFAR-100 as the ID dataset. To explore this issue, we conducted experiments with CIFAR-100 as the ID dataset and iNaturalist, Places, SUN, and Textures as the OOD datasets. Due to the presence of overlapping categories between CIFAR-100 and the OOD datasets, we manually removed the following categories from CIFAR-100: flowers (orchids, poppies, roses, sunflowers, tulips), large man-made outdoor things (bridge, castle, house, road, skyscraper), large natural outdoor scenes (cloud, forest, mountain, plain, sea), and trees (maple, oak, palm, pine, willow). This means that the ID dataset we used contains only 80 distinct categories. **The experimental results are presented in Table 2 of the Rebuttal PDF file.**
>
> In brief, we did **NOT** observe a continuous improvement in performance as the number of **total** negative labels increased. The performance trend remains an inverse-V curve, similar to what we observed with ImageNet-1k. However, the peak point of the average performance across the four OOD datasets is indeed **reached at a larger selection ratio** compared to experiments on ImageNet-1k, shifting from about 15% to approximately 50%.
>
> Theoretically, continuous performance improvement seems somewhat abnormal. When the selection ratio increases to 100%, the negative labels actually **cover all words in the lexicon without utilizing any information from the ID categories**, making it improbable to achieve optimal results. If this analysis does not fully address your concerns, we welcome you to provide more details and results of your experiments for further discussion.
>
>
> > 5.  It is suggested to follow Neglabel and refer to the introduced classes as "negative labels" instead of "OOD labels" to avoid confusion with the labels of practical OOD datasets.
>
> Thank you for your valuable suggestion! We agree that referring to the newly introduced classes as "negative labels" is indeed a more appropriate approach. We will incorporate this correction in the revised version.
>
> **In the end, we express our sincerest gratitude for your valuable suggestions and positive rating!**

---

> > ### Comment · Reviewer_gvS3 · 2024-08-08
> >
> > I'm grateful for your response. My concerns have been mostly resolved, and I now have a deeper understanding of your method. I find it simple yet effective. However, it largely builds on NegLabel (by adding adjective+noun combinations beyond separate nouns and adjectives). Therefore, I will keep my current score.

---

### Official Review · Reviewer_QWh5 · 2024-07-12

**Soundness:** 3
**Presentation:** 4
**Contribution:** 3
**Rating:** 7
**Confidence:** 4

**Summary:**

This paper presents improvements to zero-shot OOD detection methods based on pre-trained vision-language models. The study first models factors that influence the performance of existing pipelines and theoretically derive two necessary conditions for enhancing performance: expanding the OOD label candidate pool and maintaining low interdependence. Subsequently, the author analyzes why simple expansion methods do not meet these conditions and proposes constructing a conjugated semantic pool for expansion. This method meets the theoretical conditions and achieves SOTA performance on several public benchmarks.

**Strengths:**

1. The article presents reliable mathematical modeling and theoretical derivations, and the proposed method is simple and efficient.

2. The proposed method achieves considerable performance improvements over the latest methods.

3. The article features a clear structure and logical coherence, demonstrating a high standard of presentation and writing quality.

**Weaknesses:**

1. The description in the caption of Table 1 is unclear. In the experimental section, the author mentioned that the upper part of Table 1 represents traditional OOD detection methods, while the lower part pertains to detection methods based on pre-trained models like CLIP. However, this distinction is not explicitly marked in the caption, which might lead readers to misunderstand that all methods are based on the CLIP framework.

2. There is a possibility that in some cases, the chosen OOD labels from CSP may also have high similarity to ID images. The authors should provide more explanation on this point.

3. I do not quite understand the significance of the experiments in Table 5. I suggest the authors provide a more detailed discussion.

4. In Table 9, the meaning of "Size" of the corpus is not clear. Does it refer to the total number of words in the dictionary or the number of selected OOD labels?

**Questions:**

NA

**Limitations:**

Please see the weaknesses.

---

> ### Author Rebuttal · Authors · 2024-08-05
>
> We express our sincere gratitude for your valuable suggestions and the positive rating. Below, we first reiterate your comments, subsequently providing our detailed responses to each point.
>
> > 1.  The description in the caption of Table 1 is unclear. In the experimental section, the author mentioned that the upper part of Table 1 represents traditional OOD detection methods, while the lower part pertains to detection methods based on pre-trained models like CLIP. However, this distinction is not explicitly marked in the caption, which might lead readers to misunderstand that all methods are based on the CLIP framework.
>
> Thank you for the comment! We will clarify the distinction between the methods in the upper and lower parts of Table 1 in the caption to avoid any potential misunderstandings.
>
> > 2.  There is a possibility that in some cases, the chosen OOD labels from CSP may also have high similarity to ID images. The authors should provide more explanation on this point.
>
> Thank you for the insightful comment! As with the NegLabel method we followed, we used a negative label mining algorithm to minimize the similarity between selected negative labels and ID images. Specifically, we calculated the similarity between each OOD label in the CSP and the ID label space, selecting the least similar ones as the negative labels for actual use. We have provided the pseudocode for this process in the rebuttal PDF file. Although we cannot completely avoid instances where certain OOD labels may exhibit high similarity with ID images, the strategy we employed effectively decreases the likelihood of such occurrences.
>
> > 3.  I do not quite understand the significance of the experiments in Table 5. I suggest the authors provide a more detailed discussion.
>
> Thank you for your suggestion! The experiments in Table 5 primarily aim to demonstrate that, consistent with our established theory, expanding label candidates with the CSP satisfies the requirement derived in Section 3.1: concurrently enlarging the semantic pool size M and the expected activation probability q_2 of OOD labels. Specifically:
>
> Since the superclasses used in constructing the CSP typically include broad semantic objects, the property clusters encompass samples from numerous potential OOD categories. Therefore, their centers have much higher expected probabilities of being activated by OOD samples, which brings an increase in q_2. In Table 5, we present the expected softmax scores for a single OOD label from both the original semantic pool and the CSP. These scores, averaged across OOD samples, serve as an approximation of q_2, which is defined as the expected probability of OOD labels being activated by OOD samples. Table 5 reveals that the average score of our CSP across four OOD datasets is distinctly higher than that of the original pool, indicating that this expansion leads to an increase in q_2.
>
> > 4.  In Table 9, the meaning of "Size" of the corpus is not clear. Does it refer to the total number of words in the dictionary or the number of selected OOD labels?
>
> Thank you for your question! Since we used nouns from the lexicon to construct the original semantic pool and adjectives to build the conjugated semantic pool, the "Size" in Table 9 refers to the total number of nouns and adjectives in the lexicon, not the number of selected OOD labels.
>
> **In the end, we express our sincerest gratitude for your time and consideration!**

---

> > ### Comment · Reviewer_QWh5 · 2024-08-12
> > **Official Comment by Reviewer QWh5**
> >
> > Thanks to the authors for the response. I am happy that most of my concerns have been addressed and I have decided to keep the score.

---

### Official Review · Reviewer_pEkd · 2024-07-14

**Soundness:** 2
**Presentation:** 3
**Contribution:** 2
**Rating:** 4
**Confidence:** 4

**Summary:**

The paper proposes a method for zero-shot out-of-distribution (OOD) detection using an expanded semantic pool of modified superclass names. By leveraging a pre-trained vision-language model, the approach aims to improve OOD classification performance by ensuring low mutual dependence among selected OOD labels. This method outperforms existing techniques by 7.89% in FPR95, highlighting its effectiveness in handling OOD detection tasks.

**Strengths:**

1. The theoretical analysis of expanding NegLabel using the u-function is very clear and intuitive.
2. The experiments are conducted very comprehensively.

**Weaknesses:**

1. The method proposed in this paper lacks innovation; its main framework and basic performance are entirely derived from NegLabel.
2.The preliminary section lacks clear references to the original theory of NegLabel and the proofs in the appendix.
3.The description of the proposed method is very concise, which makes it unclear for readers to understand the details of the method design. It is recommended to provide pseudocode for the algorithm to clarify the method's design.

**Questions:**

see weakness

**Limitations:**

YES

---

> ### Author Rebuttal · Authors · 2024-08-05
>
> We express our sincere gratitude for your constructive comments. Below, we first reiterate your comments, subsequently providing our detailed responses to each point.
>
> > 1. The method proposed in this paper lacks innovation; its main framework and basic performance are entirely derived from NegLabel.
>
> Thank you for your comments! While we utilized the existing SOTA method, NegLabel, as our primary framework, we believe that our work provides meaningful theoretical and performance advancements, showcasing unique innovation. We sincerely hope the following explanation addresses your concerns, and we welcome any further discussion on this matter.
>
> Regarding the methodological framework, we have discussed the theoretical shortcomings in NegLabel's derivation (L1030-1042) : NegLabel undertakes a rudimentary theoretical analysis of the correlation between OOD detection performance and the quantity of adopted potential labels, concluding that an increase in selected labels correlates with enhanced performance. However, this conclusion contradicts the observed actual trend. The contradiction arises from that NegLabel simply assumes a constant higher similarity between OOD labels and OOD images compared to ID images, **neglecting that this similarity discrepancy originates from the strategy of reverse-order selection of OOD labels based on their similarity to the ID label space**. As the set of selected OOD labels transitions from "*a small subset of labels with the lowest similarity to the entire ID label space*" to "*the whole semantic pool, which is unrelated to the setting of ID and OOD labels*", the discrepancy in similarity of ID images to OOD labels versus OOD images to OOD labels will progressively diminish until it disappears. **Reviewer p3ao also comments that** "the supplement to the shortcomings of the NegLabel theory is very enlightening".
>
> Building on this insight, we incorporated this dynamic process into our analysis and optimize the performance modeling of NegLabel, leading to the derivation of the unique mathematical model presented in Section 3.1. With a series of derivation, this model establishes the conditions for performance improvement through semantic pool expansion: an unequivocal strategy for performance enhancement requires **concurrently increasing the semantic pool size and the expected activation probability of OOD labels** and **ensuring low mutual dependence among the activations of selected OOD labels**.
>
> In Section 3.2, based on the optimized mathematical model, we analyzed why simple lexicon expansion fails to yield further performance improvements. In Section 3.3, we proposed a novel method for constructing a conjugate semantic pool, expanding the semantic pool in a manner that satisfies the theoretical conditions mentioned above, and achieved satisfactory performance gains. Therefore, **all the derivations and design choices detailed in Methodology are contributions we have made to this framework**.
>
> In terms of basic performance, while NegLabel, as the current SOTA method, indeed demonstrates strong performance, our approach has achieved considerable improvements over NegLabel on the standard ImageNet-1k OOD detection benchmark, with a **1.55% increase in AUROC and a 7.89% reduction in FPR95**. Furthermore, our method significantly outperforms NegLabel in various scenarios, including hard OOD detection tasks (*Table 2*), different CLIP models (*Table 3*), different ID datasets (*Table 7*), different corpus sources (*Table 9*), and different VLM architectures (*Table 10*). The effectiveness of our approach has been validated through extensive experiments. **The strong performances we achieved stem from our theoretical analysis and the proposed conjugated semantic pool**, rather than simply from the use of NegLabel.
>
> > 2. The preliminary section lacks clear references to the original theory of NegLabel and the proofs in the appendix.
>
> Thank you for the valuable comment! We originally structured the paper to clearly differentiate contributions, with NegLabel's contributions discussed in the **Preliminary** section and our own in **Methodology**. We apologize for any ambiguity and will clarify this in the revised version.
>
> In Preliminary, we have cited NegLabel at **L99 and L104**, to introduce **NegLabel's methodology** (L97-103) and its **theoretical performance modeling** (L104-124), respectively, summarizing the contributions made by NegLabel. We will make the references more explicit in the revised version.
>
> In Appendix A.1 and A.2, to ensure clarity and proper acknowledgment, we will add the following statement: “**The proof in this part is adapted from the appendix of [24]**”.
>
> Additionally, we have cited NegLabel [24] a total of **15** times, with its name, "NegLabel," appearing **20** times, throughout the paper. This should make it clear that our method has strong relevance with the NegLabel framework, upon which we have introduced further innovations. If there are any aspects where the contributions of NegLabel have not been clearly articulated, we would greatly appreciate the opportunity for further discussion and are fully supportive of providing any necessary clarifications.
>
> > 3. The description of the proposed method is very concise, which makes it unclear for readers to understand the details of the method design. It is recommended to provide pseudocode for the algorithm to clarify the method's design.
>
> Thank you for the constructive comment! Following the suggestion, we have **provided the pseudocode of our algorithm in the rebuttal PDF file** as Algorithm 1, and we will incorporate it, along with a more detailed version of the method design, into the appendix in the revised version. If there are any aspects of the algorithmic process that remain unclear, we would be more than happy to engage in further discussion.
>
> **Finally, we extend our sincerest thanks for all your time and consideration!**

---

### Author Rebuttal · Authors · 2024-08-05

We thank all the reviewers for their thorough reading of our work and the high-quality feedback they provided. Their comments have been immensely beneficial in enhancing the quality of our manuscript and deepening our own understanding of this field. We have uploaded detailed, point-by-point responses for each reviewer, which we believe address all the concerns raised. Additionally, we have provided the requested algorithm pseudocode and extra experimental results in the uploaded rebuttal PDF file, as referenced in our responses to the corresponding reviewers

We look forward to engaging in further discussion and exchange with the reviewers during the next stage of the review process. Finally, we extend our sincerest thanks to all the reviewers for their time and thoughtful consideration!

---

### Author Response · Authors · 2024-08-14
**Summary of reviews and rebuttal**

Dear Reviewers, ACs, and SACs,

We would like to express our sincerest gratitude to the reviewers, ACs, and SACs for the time and effort invested in reviewing our work. As the review discussion period comes to a close, we would like to provide a brief summary of the main strengths, weaknesses, and our rebuttal.

**Strengths**:
1. The theoretical analysis/motivation is clear and intuitive (pEkd, gvS3, KrTH), and the mathematical modeling is reliable/robust (QWh5, KrTH).
2. The analysis of the shortcomings/limitations of NegLabel/existing methods is enlightening (p3ao, KrTH), and the method design is simple/straightforward and effective (gvS3, QWh5).
3. The experiments are comprehensive (pEkd, p3ao, KrTH) , and the performance improvement is considerable/significant (gvS3, KrTH, QWh5). SOTA is achieved (gvS3,KrTH).
4. The paper is well-written/has high standard of presentation and writing quality (p3ao, QWh5).

The above strengths were the ones appreciated by more than one reviewers, and we are deeply grateful for their recognition and positive feedback.

**Weaknesses/Questions**:

1. This work is primarily based on the framework of the SOTA method NegLabel (p3ao, pEkd).

**Rebuttal**: This work indeed builds on NegLabel. However, leveraging open vocabulary and pre-trained models is a natural and common paradigm for zero-shot tasks. Based on the existing framework, our work identifies key shortcomings and achieves much superior performance through our derived theoretical guidance and straightforward modifications to the semantic pool, showcasing unique innovation.

2. The description of the algorithm (pEkd) / table1 caption and table5 significance (QWh5) / technical details (gvS3, p3ao) are too concise.

**Solution**: We have provided a more detailed introduction and the pseudo code in the rebuttal, which will also be added to the revised version.

3. Questions related to synonyms in the semantic pool (KrTH, p3ao).

**Answer**: We have carefully explained the effect of including synonyms and how we handle them, which have addressed the concerns of reviewers. Please refer to the rebuttal for details.

The above weaknesses and questions were the ones raised by more than one reviewers. Besides, all reviewers who responded agreed that our rebuttal addresses their concerns.

Finally, thank you again for your valuable time and expertise throughout this process. Should you have any further suggestions or comments, we are more than willing to consider them and make the necessary adjustments.

Warm regards,

Authors of Submission 5027

---

### Decision · Program_Chairs · 2024-09-25

**Decision:**

Accept (poster)

**Comment:**

This paper proposes a novel method to build a Conjugated Semantic Pool to improve OOD detection by pretrained vision and language model. Concretely, the paper identifies two key factors to build the semantic pool: 1) increasing the activation probability of selected OOD labels, and  2) maintaining low interdependence among the chosen OOD labels. The proposed approach has sound theoretical support and comprehensive experiments to show the significant improvement over prior works e.g. NegLabel.

All reviewers converge on high scores 7, 6, 7, 6 to accept the paper (except one with a score of 4). The reviewers like the clear and intuitive theoretical justification (pEkd, gvS3, KrTH), and analysis of the shortcomings of existing methods (p3ao, KrTH). In addition, the experiments are thorough (pEkd, p3ao, KrTH), and demonstrate significant gains over current SoTA (gvS3, KrTH, QWh5). After considering the author reviews, rebuttal and the paper, I'd recommend acceptance of the paper.